# ParB spreading on DNA requires cytidine triphosphate in vitro

**Adam SB Jalal, Ngat T Tran, Tung BK Le***

Department of Molecular Microbiology, John Innes Centre, Norwich, United Kingdom

**Abstract** In all living organisms, it is essential to transmit genetic information faithfully to the next generation. The SMC-ParAB-*parS* system is widely employed for chromosome segregation in bacteria. A DNA-binding protein ParB nucleates on *parS* sites and must associate with neighboring DNA, a process known as spreading, to enable efficient chromosome segregation. Despite its importance, how the initial few ParB molecules nucleating at *parS* sites recruit hundreds of further ParB to spread is not fully understood. Here, we reconstitute a *parS*-dependent ParB spreading event using purified proteins from *Caulobacter crescentus* and show that CTP is required for spreading. We further show that ParB spreading requires a closed DNA substrate, and a DNA-binding transcriptional regulator can act as a roadblock to attenuate spreading unidirectionally in vitro. Our biochemical reconstitutions recapitulate many observed in vivo properties of ParB and opens up avenues to investigate the interactions between ParB-*parS* with ParA and SMC.

## Introduction

Faithful chromosome segregation is essential in all domains of life if daughter cells are each to inherit the full set of genetic information. The SMC-ParAB-*parS* complex is widely employed for chromosome segregation in bacteria (*Donczew et al., 2016*; *Fogel and Waldor, 2006*; *Gruber and Errington, 2009*; *Harms et al., 2013*; *Ireton et al., 1994*; *Jakimowicz et al., 2002*; *Kawalek et al., 2018*; *Lin and Grossman, 1998*; *Livny et al., 2007*; *Mohl et al., 2001*; *Sullivan et al., 2009*; *Tran et al., 2018*; *Wang et al., 2017*). The centromere *parS* is the first DNA locus to be segregated following chromosome replication (*Lin and Grossman, 1998*; *Livny et al., 2007*; *Toro et al., 2008*; *Lagage et al., 2016*). ParB specifically nucleates on *parS* before spreading outwards to the flanking DNA and bridges/cages DNA together to form a nucleoprotein network in vivo (*Breier and Grossman, 2007*; *Murray et al., 2006*; *Taylor et al., 2015*; *Graham et al., 2014*; *Sanchez et al., 2015*; *Debaugny et al., 2018*; *Broedersz et al., 2014*; *Funnell, 2016*). This nucleoprotein complex recruits SMC to disentangle and organize replicated DNA (*Gruber and Errington, 2009*; *Sullivan et al., 2009*; *Wang et al., 2017*; *Tran et al., 2017*; *Minnen et al., 2011*). ParB-*parS* also interacts with an ATPase ParA to power the segregation of replicated chromosomes (*Lim et al., 2014*; *Vecchiarelli et al., 2014*; *Vecchiarelli et al., 2012*; *Hwang et al., 2013*; *Leonard et al., 2005*). Engineered strains harboring a nucleation-competent but spreading-defective mutant of *parB* are either unviable (*Mohl et al., 2001*) or have elevated number of anucleate cells (*Harms et al., 2013*; *Kawalek et al., 2018*; *Lin and Grossman, 1998*; *Lagage et al., 2016*; *Jecz et al., 2015*; *Attaiech et al., 2015*; *Yu et al., 2010*; *Lee and Grossman, 2006*). Despite the importance of spreading for proper chromosome segregation, the mechanism by which a few *parS*-bound ParB can recruit hundreds more ParB molecules to the vicinity of *parS* to assemble a high-molecular-weight nucleoprotein complex is not fully understood.

Since the first report in 1995 (*Lynch and Wang, 1995*), ParB spreading has been observed in vivo by chromatin immunoprecipitation in multiple bacterial species (*Tran et al., 2018*; *Lagage et al., 2016*; *Breier and Grossman, 2007*; *Murray et al., 2006*; *Graham et al., 2014*; *Rodionov et al.,*

*For correspondence:
tung.le@jic.ac.uk

Competing interests: The authors declare that no competing interests exist.

*1999*). The nucleation of ParB on *parS* has also been demonstrated in vitro (*Harms et al., 2013*; *Mohl et al., 2001*; *Breier and Grossman, 2007*; *Murray et al., 2006*; *Sanchez et al., 2015*; *Chen et al., 2015*; *Surtees and Funnell, 2001*; *Ah-Seng et al., 2013*); however, *parS*-dependent ParB spreading has resisted biochemical reconstitution (*Taylor et al., 2015*; *Graham et al., 2014*; *Fisher et al., 2017*; *Madariaga-Marcos et al., 2019*). Unsuccessful attempts to reconstitute spreading in vitro suggests that additional factors might be missing. Recently, works by *Soh et al. (2019)* and *Osorio-Valeriano et al. (2019)* on *Bacillus subtilis* and *Myxococcus xanthus* ParB, respectively, showed that ParB binds and hydrolyzes cytidine triphosphate (CTP) to cytidine diphosphate (CDP), and that CTP modulates the binding affinity of ParB to *parS* (*Osorio-Valeriano et al., 2019*; *Soh et al., 2019*). A co-crystal structure of *Bacillus* ParB with CDP and that of a *Myxococcus* ParB-like protein (PadC) with CTP showed CTP binding promotes a new dimerization interface between N-terminal domains of ParB subunits (*Osorio-Valeriano et al., 2019*; *Soh et al., 2019*). Crucially, *Soh et al. (2019)* showed by single-molecule imaging and cross-linking assays that *Bacillus* ParB, in the presence of CTP, forms a self-loading protein clamp at *parS* and slides away to spread to neighboring DNA (*Soh et al., 2019*). While reproducing a key result from *Easter and Gober (2002)* that showed *Caulobacter crescentus* ParA-ATP dissociated pre-bound ParB from *parS* (*Easter and Gober, 2002*), we found that CTP could also modulate the nucleation of *Caulobacter* ParB on *parS*. Personal communication with Stephan Gruber and the recent work by *Osorio-Valeriano et al. (2019)* and *Soh et al. (2019)* encouraged us to take steps further to investigate the role of CTP for ParB spreading in *Caulobacter crescentus*.

Here, we reconstitute a *parS*-dependent ParB spreading on DNA in real-time, using a label-free purified protein from *Caulobacter crescentus*. Consistent with pioneering works by *Soh et al. (2019)* and *Osorio-Valeriano et al. (2019)*, we confirm that CTP regulates ParB-DNA interaction. We further provide evidence that the accumulation of *Caulobacter* ParB requires a closed DNA substrate, and that a DNA-binding transcription factor, TetR, can act as a roadblock to attenuate ParB accumulation unidirectionally in vitro. Our real-time and label-free reconstitution has successfully recapitulated many well-known aspects of ParB behaviors in vivo and might open up avenues to investigate further the roles of the ParB-*parS* nucleoprotein complex in ensuring faithful chromosome segregation.

## Results

### CTP reduces the nucleation of *Caulobacter* ParB on *parS*

*Easter and Gober (2002)* reported that ATP-bound *Caulobacter* ParA dissociated ParB from *parS* (*Easter and Gober, 2002*); however, the authors did not control for the effect of ATP alone on ParB-*parS* binding. NTPs are highly negatively charged and could have affected protein-DNA interactions by binding non-specifically to the often positively charged DNA-binding domain. To determine if ATP or other NTP alone affects ParB-*parS* interaction, we attached a linear 20 bp biotinylated *parS* DNA to a streptavidin-coated probe to measure the bio-layer interference (BLI) (*Figure 1A*). BLI assay monitors wavelength shifts (responses) resulting from changes in the optical thickness of the probe surface during association or dissociation of the analyte (see Materials and Methods). We monitored in real-time interactions between immobilized *parS* DNA and purified *Caulobacter* ParB or a premix of ParB + NTP (*Figure 1B*). Consistent with previous reports (*Tran et al., 2018*; *Figge et al., 2003*), *Caulobacter* ParB bound site-specifically to *parS* but not to non-cognate DNA (*Figure 1—figure supplement 1*). In the presence of ATP, GTP, or UTP, we observed a small reduction in ParB-*parS* binding at steady state regardless of whether $Mg^{2+}$ was included in binding buffer or not (*Figure 1B* and *Figure 1—figure supplement 2*), suggesting that *Caulobacter* ParB is slightly sensitive to highly negatively charged compounds or to counter-ions ($Na^+$) in NTP solutions. However, we noted that CTP had a pronounced effect on ParB-*parS* interaction, specifically in the presence of $Mg^{2+}$ (*Figure 1B* and *Figure 1—figure supplement 2*). An increasing concentration of CTP (but not CMP or CDP) gradually reduced the binding of ParB to *parS* (*Figure 1—figure supplement 3*). In contrast, neither CTP nor other NTPs affected the binding of another protein-DNA pair, for example, the well-characterized TetR-*tetO* interaction (*Saenger et al., 2000*; *Figure 1C*). On closer inspection, we noted that ParB + CTP slowly dissociated from *parS* even before the probe was returned to a protein-free buffer (a gradual downward slope between 30th and 150th sec,

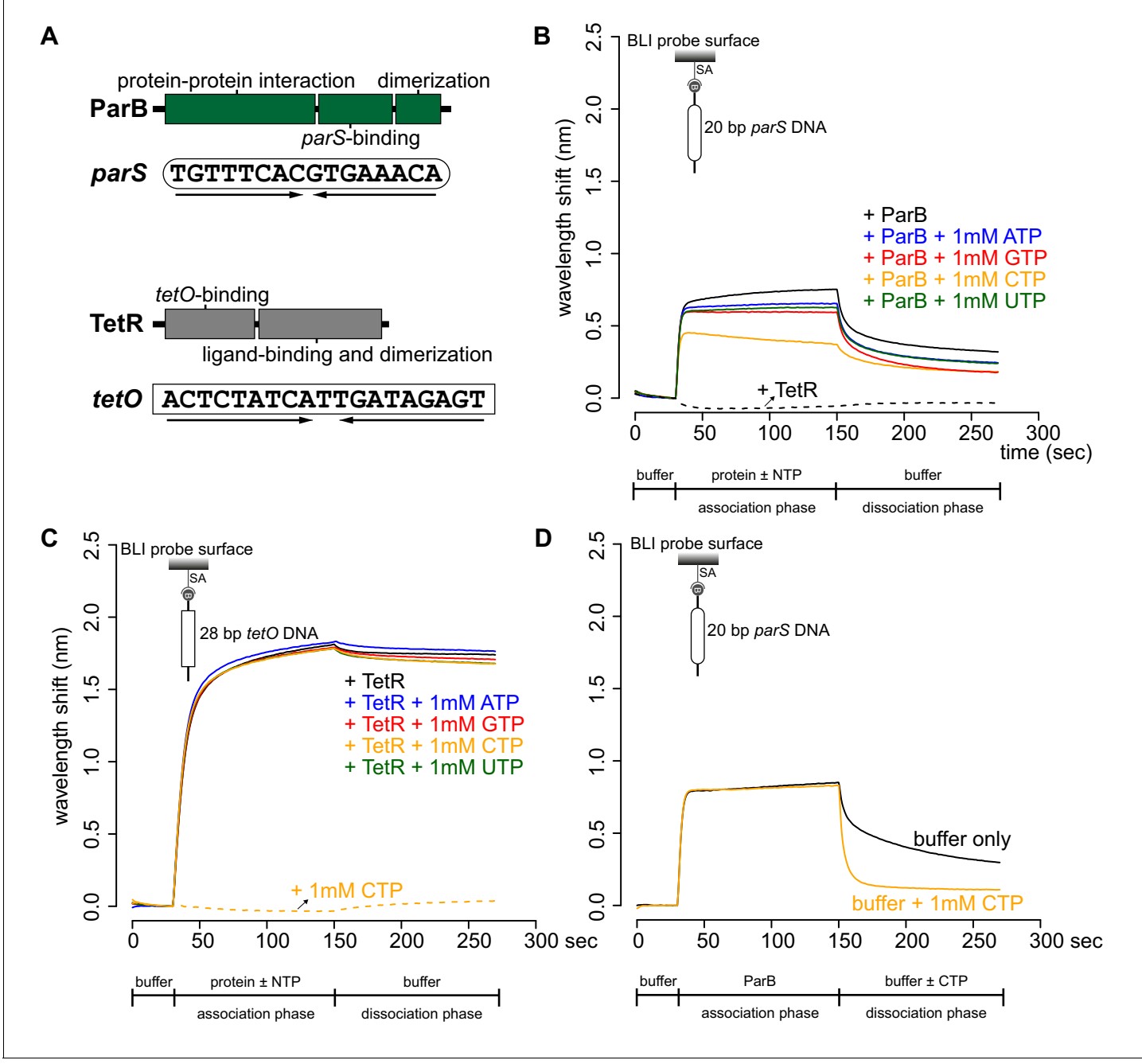

**Figure 1.** CTP reduces the nucleation of *Caulobacter* ParB at *parS*. (**A**) The domain architecture of ParB (dark green) and TetR (grey), and their respective DNA-binding sites *parS* and *tetO*. Convergent arrows below DNA-binding sites indicate that *parS* and *tetO* are palindromic. (**B**) Bio-layer interferometric (BLI) analysis of the interaction between a premix of 1 μM ParB-His6 dimer ± 1 mM NTP and a 20 bp DNA duplex containing *parS*. Biotinylated DNA fragments were immobilized onto the surface of a Streptavidin (SA)-coated probe (See Materials and methods). The BLI probe was dipped into a buffer only solution (0–30 s), then to a premix of protein ± NTP (30–150 s: association phase), and finally returned to a buffer only solution (150–270 s: dissociation phase). Sensorgrams were recorded over time. BLI analysis of the interaction between 1 μM TetR-His6 and a 20 bp *parS* probe was also recorded (a negative control). (**C**) BLI analysis of the interaction between a premix of 1 μM TetR-His6 ± 1 mM NTP and a 28 bp DNA duplex containing *tetO*. BLI analysis of the interaction between 1 mM CTP and a 28 bp *tetO* probe was also recorded (a negative control). (**D**) BLI analysis of the interaction between 1 μM *Caulobacter* ParB-His6 (without CTP) and a 20 bp *parS* DNA. For the dissociation phase, the probe was returned to a buffer only or buffer supplemented with 1 mM CTP. All buffers used for experiments in this figure contained Mg2+. Each BLI experiment was triplicated and a representative sensorgram was presented.

The online version of this article includes the following source data and figure supplement(s) for figure 1:

**Source data 1.** Data used to generate *Figure 1*.

*Figure 1 continued on next page*

*Figure 1 continued*

**Figure supplement 1.** ParB and TetR bind specifically to their cognate-binding sites *parS* and *tetO*, respectively.
**Figure supplement 1—source data 1.** Data used to generate *Figure 1—figure supplement 1*.
**Figure supplement 2.** BLI analysis of the interaction between purified *Caulobacter* ParB and NTP in buffers lacking $Mg^{2+}$.
**Figure supplement 2—source data 1.** Data used to generate *Figure 1—figure supplement 2*.
**Figure supplement 3.** BLI analysis of the interaction between purified *Caulobacter* ParB and cytidine mono-, di-, or triphosphate.
**Figure supplement 3—source data 1.** Data used to generate *Figure 1—figure supplement 3*.

*Figure 1B*), suggesting that CTP facilitated ParB removal from a 20 bp *parS* DNA. To investigate further, we monitored the dissociation rates of pre-bound CTP-free ParB-*parS* complexes after probes were returned to a protein-free buffer with or without CTP, we found ParB dissociating ~seven times faster in buffer with CTP than in buffer only solution (*Figure 1D*). Given the short length of a 20 bp *parS* DNA duplex that has only sufficient room for nucleation, our results suggest that CTP might decrease ParB nucleation on *parS* or liberates pre-bound ParB from *parS* site.

## CTP facilitates ParB accumulation on a closed DNA substrate

Next, we investigated the effect of CTP on ParB-DNA interaction by employing a longer 169 bp *parS*-containing DNA fragment that has been labeled at both 5' ends with biotin (*Figure 2A*). Immobilizing a dual biotin-labeled DNA on a streptavidin-coated BLI probe created a DNA substrate where both ends were blocked (a closed DNA) (*Onn and Koshland, 2011*; *Figure 2—figure supplement 1*). We monitored the interactions between immobilized DNA and purified *Caulobacter* ParB in the presence or absence of NTP. In the absence of NTP, we observed the usual nucleation event on *parS* with 1 μM ParB protein (*Figure 2A*). We noted that the BLI signal was not as high as with a 20 bp *parS* probe (*Figure 2A*) due to a less efficient immobilization of a longer DNA fragment on the BLI probe. Premixing ATP, GTP, or UTP with ParB did not change the sensorgrams markedly (*Figure 2A*). However, the addition of CTP significantly increased the response by ~12 fold (*Figure 2A* and *Figure 2—figure supplement 2A*), suggesting that more ParB associated with a 169 bp *parS* probe at steady state than by nucleation at *parS* alone. We observed that DNA-bound ParB was salt sensitive and dissociated readily to the solution when the BLI probe was returned to a low-salt protein-free buffer without CTP (*Figure 2A*, dissociation phase). However, the dissociation of pre-bound ParB-CTP from DNA was slowed down by ~fivefold if the probe was returned to a buffer supplemented with CTP (*Figure 2—figure supplement 2B*). The effect on the BLI response was not seen if $Mg^{2+}$ was omitted (*Figure 2—figure supplement 2C*), neither did we observe an equivalent increase in response when a 169 bp dual biotin-labeled DNA containing a scrambled *parS* was employed instead (*Figure 2A*). Furthermore, we observed that a nucleation-competent but spreading-defective *Caulobacter* ParB (R104A) (*Tran et al., 2018*) mutant did not respond to the addition of CTP to the same extent as ParB (WT) (*Figure 2B*). Our results suggest that CTP is required for the increase in *parS*-dependent ParB accumulation in vitro. Lastly, we performed BLI experiments for eight additional chromosomal ParB proteins from a diverse set of bacterial species and consistently observed the specific effect of CTP on enhancing ParB association with a closed DNA in vitro (*Figure 2—figure supplement 3*). It is most likely that ParB-CTP interaction with DNA is conserved among ParB orthologs.

To independently verify the BLI data, we performed an in vitro pull-down of purified His-tagged *Caulobacter* ParB (*Figure 2C*). Streptavidin-coated paramagnetic beads were incubated with 2.8 kb dual biotin-labeled DNA fragments containing either *parS* or scrambled *parS* sites. Again, a dual biotin-labeled DNA formed a closed substrate on the surface of the beads. DNA-coated beads were incubated with purified *Caulobacter* ParB either in the presence or absence of NTP before being pulled down magnetically. Pulled-down ParB was released from beads and their protein level was analyzed by an α-His$_6$ immunoblot (*Figure 2C*). We found ~13–15 fold more pulled-down ParB when CTP was included (*Figure 2C*). No enrichment was observed when scrambled *parS*-coated beads were used, confirming that the extensive in vitro association of ParB with DNA is dependent on *parS* (*Figure 2C*). Also, consistent with the BLI experiments, no further enrichment of ParB was seen when ATP, GTP or UTP was included (*Figure 2C*). Furthermore, a ParB (R104A) variant was not enriched in our pull-down assay regardless of whether CTP was present or not (*Figure 2C*). Altogether, our

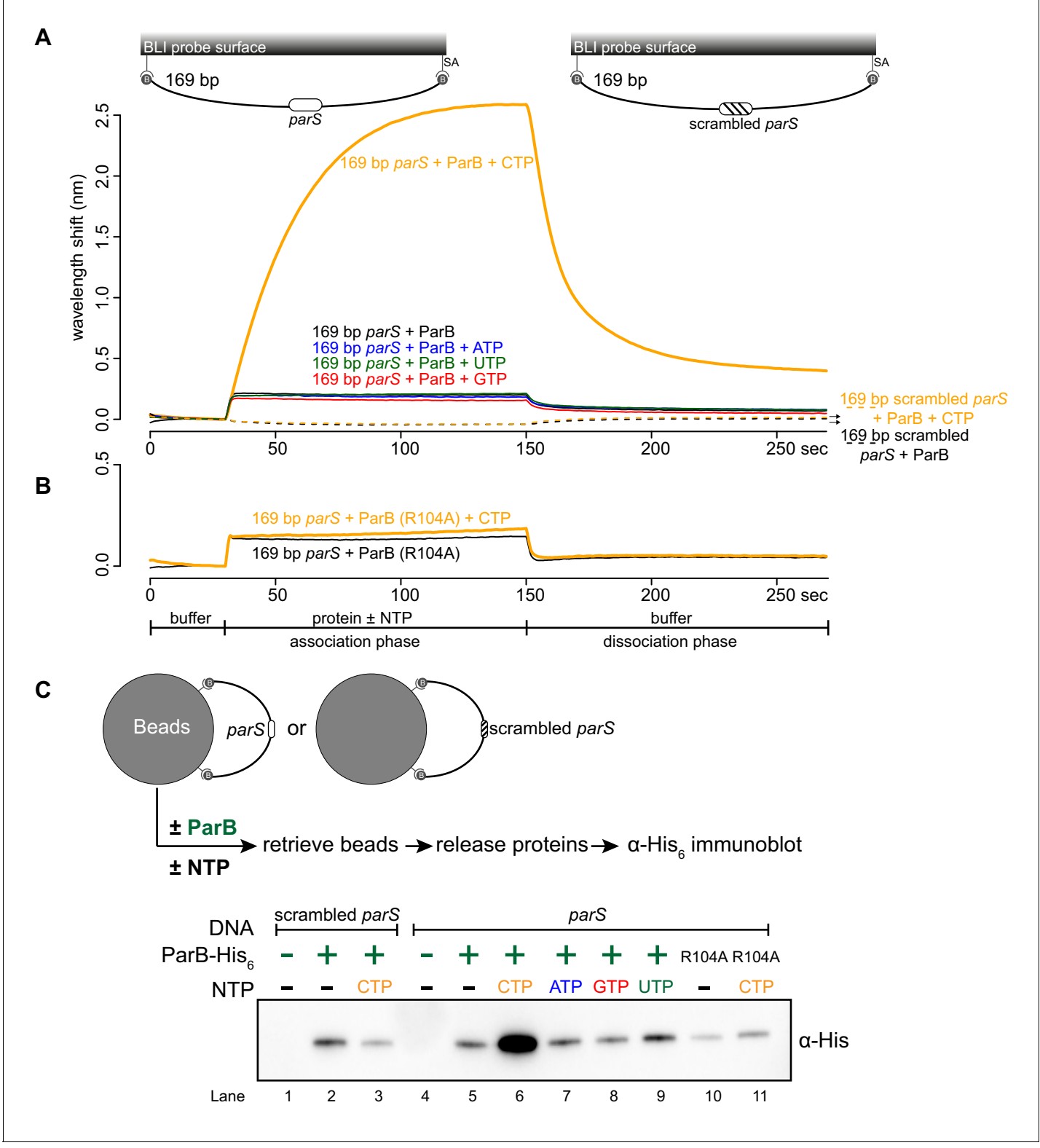

**Figure 2.** CTP facilitates ParB association with a closed DNA substrate beyond nucleation. (**A**) BLI analysis of the interaction between a premix of 1 µM *Caulobacter* ParB-His$_6$ ± 1 mM NTP and a 169 bp dual biotin-labeled DNA containing a *parS* or a scrambled *parS* site. Interactions between a dual biotinylated DNA and streptavidin (SA)-coated probe created a DNA substrate where both ends were blocked (a closed DNA substrate) (see the schematic of the BLI probes above the sensorgram). (**B**) Interactions between a nucleation-competent but spreading-defective ParB (R104) variant with a 169 bp *parS* DNA fragment in the presence or absence of CTP were also recorded. Each BLI experiment was triplicated and a representative *Figure 2 continued on next page*

*Figure 2 continued*

sensorgram was presented. (**C**) A schematic of the pull-down assay and immunoblot analysis of pulled-down *Caulobacter* ParB-His$_6$. The length of bound DNA is ~2.8 kb. Beads were incubated with ParB protein for five minutes before being pulled down magnetically. All buffers used for experiments in this figure contained Mg$^{2+}$.

The online version of this article includes the following source data and figure supplement(s) for figure 2:

**Source data 1.** Data used to generate *Figure 2*.
**Figure supplement 1.** Dual biotin-labeled DNA fragments form a closed substrate on the surface of the BLI probe.
**Figure supplement 1—source data 1.** Data used to generate *Figure 2—figure supplement 1*.
**Figure supplement 2.** CTP-Mg$^{2+}$ enhances ParB accumulation on a closed DNA substrate.
**Figure supplement 2—source data 1.** Data used to generate *Figure 2—figure supplement 2*.
**Figure supplement 3.** CTP facilitates the association of eight ParB orthologs with DNA.
**Figure supplement 3—source data 1.** Data used to generate *Figure 2—figure supplement 3*.

results suggest that a *parS*-dependent accumulation of ParB on a closed DNA substrate requires CTP.

## A closed DNA substrate is required for an increased ParB association with DNA

Next, we investigated whether a DNA substrate with a free end (an open DNA) can also support ParB accumulation in vitro. The 169 bp dual biotin-labeled DNA was designed with unique *Bam*HI and *Eco*RI recognition sites flanking the *parS* site (*Figure 3A*). To generate an open end on DNA, we immerged the DNA-coated probes in buffers contained either *Bam*HI or *Eco*RI (*Figure 3A–C* and *Figure 3—figure supplement 1*). Before restriction enzyme digestion, we again observed an enhanced ParB association with a closed DNA substrate in the presence of CTP (*Figure 3A*). After digestion by either *Bam*HI or *Eco*RI, the inclusion of CTP had no effect on the BLI response, indicating that ParB did not accumulate on an open DNA substrate in vitro (*Figure 3B–C*). Our conclusion was further supported by results from an experiment in which we added *Bam*HI after ParB + CTP and a closed DNA substrate were preincubated together for 120 s (*Figure 3—figure supplement 2*). Here, in the presence of *Bam*HI, ParB binding to DNA reduced gradually over 30 min, while it was unaffected if heat-inactivated *Bam*HI was employed instead (*Figure 3—figure supplement 2*). Lastly, consistent with BLI experiments, our pull-down assay also showed that ParB-CTP failed to accumulate when a 2.8 kb dual biotin-labeled DNA was linearized by *Hind*III digestion (*Figure 3D*).

Next, we wondered if a tight protein-DNA binding could cap the open end of DNA, thereby mimicking a closed DNA substrate and restoring ParB accumulation. To investigate this possibility, we constructed a 170 bp dual biotin-labeled DNA fragment that contains a single *parS* site, a *tetO* operator, and flanking restriction enzyme recognition sites for *Eco*RI and *Bam*HI (*Figure 4A*). With this closed DNA substrate, we observed an enhanced ParB association with DNA in the presence of CTP (*Figure 4A*). Again, we generated a free DNA end via restriction enzyme digestion. Consistent with previous experiments with a restricted 169 bp DNA probe, the addition of ParB + CTP had no effect on the BLI response (*Figure 4B*). However, it can be partially rescued by incubating a *Bam*HI-restricted DNA probe with a premix of ParB + CTP + TetR (*Figure 4B*). We reason that TetR binding at *tetO* capped the open DNA end, essentially generated a closed DNA substrate. Our conclusion was further supported by results from an experiment in which a premix of ParB + CTP + TetR was tested against an *Eco*RI-restricted DNA instead (*Figure 4C*). Here, we did not observe an enhanced association of ParB with DNA even when TetR was included, most likely because of a persistent open DNA end that could not be blocked by TetR-*tetO* binding (*Figure 4C*). Finally, we observed that the TetR could attenuate ParB-CTP accumulation on a closed DNA substrate (magenta line vs. orange line, *Figure 4A*). This blocking effect is specific to TetR-*tetO* binding since the addition of anhydrotetracycline (ahTc), a negative effector of TetR (*Saenger et al., 2000*), allowed ParB-CTP to regain its accumulation on DNA (*Figure 4—figure supplement 1*). Overall, the ability of a DNA-bound TetR to act as a roadblock in vitro is consistent with previous ChIP-seq data that showed DNA-binding proteins or RNA polymerases could block or attenuate ParB spreading unidirectionally in vivo (*Breier and Grossman, 2007*; *Murray et al., 2006*; *Sanchez et al., 2015*; *Rodionov et al., 1999*).

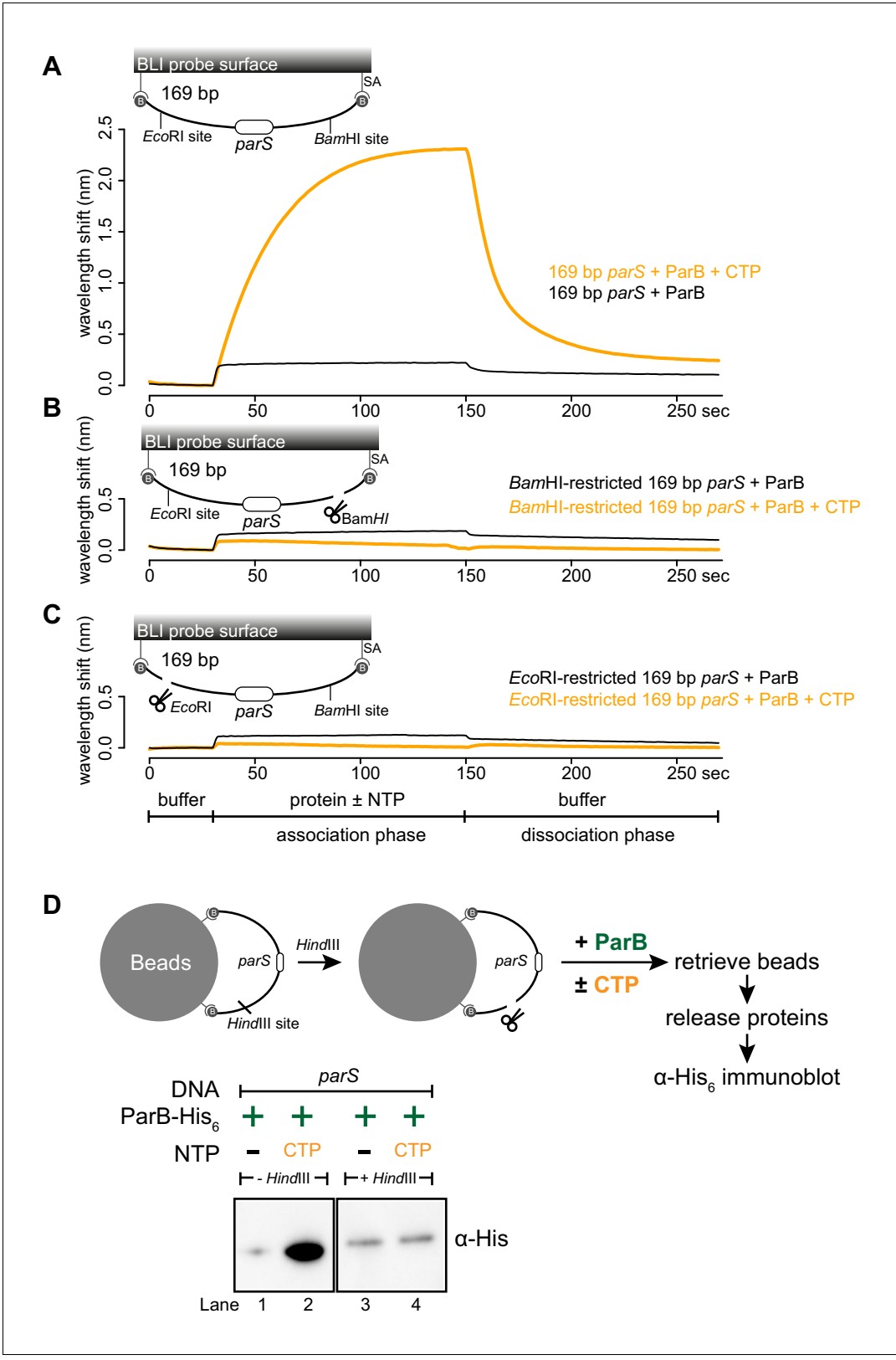

**Figure 3.** A closed DNA substrate is required for an increased association of ParB with DNA. (**A**) BLI analysis of the interaction between a premix of 1 μM *Caulobacter* ParB-His$_6$ ± 1 mM CTP and a 169 bp dual biotin-labeled *parS* DNA. (**B**) Same as panel A but immobilized DNA fragments have been restricted with *Bam*HI before BLI analysis. (**C**) Same as panel A but immobilized DNA fragments have been restricted with *Eco*RI before BLI analysis.

*Figure 3 continued on next page*

*Figure 3 continued*

Schematic of DNA fragments with the relative positions of *parS* and restriction enzyme recognition sites are shown above the sensorgram. Each BLI experiment was triplicated and a representative sensorgram was presented. (D) A schematic of the pull-down assay and immunoblot analysis of pulled-down *Caulobacter* ParB-His$_6$. For lanes 3 and 4, DNA-bound beads were incubated with *Hind*III to linearize bound DNA. Samples in lanes 1–4 were loaded on the same gel, the immunoblot was spliced together for presentation purposes. All buffers used for experiments in this figure contained Mg$^{2+}$.

The online version of this article includes the following source data and figure supplement(s) for figure 3:

**Source data 1.** Data used to generate *Figure 3*.
**Figure supplement 1.** Restriction enzymes linearize dual biotin-labeled DNA fragments on the surface of the BLI probe.
**Figure supplement 1—source data 1.** Data used to generate *Figure 3—figure supplement 1*.
**Figure supplement 2.** Linearization of a closed DNA substrate by *Bam*HI liberates pre-bound ParB from DNA.
**Figure supplement 2—source data 1.** Data used to generate *Figure 3—figure supplement 2*.

## *parS* DNA increases the CTP binding and hydrolysis rate of *Caulobacter* ParB

Recently, *Myxococcus* and *Bacillus* ParB were shown to bind and hydrolyze CTP (*Osorio-Valeriano et al., 2019*; *Soh et al., 2019*). Our in vitro results so far also hint at CTP binding directly to *Caulobacter* ParB. By employing a membrane-spotting assay (DRaCALA), we showed that *Caulobacter* ParB binds to radiolabeled CTP in the presence of *parS* DNA (*Figure 5A*). An excess of unlabeled CTP, but no other NTPs, could compete with radioactive CTP for binding to *Caulobacter* ParB (*Figure 5B*), suggesting that *Caulobacter* ParB does not bind other NTPs. The CTP binding of ParB was reduced when a non-cognate DNA site (*NBS*) (*Wu and Errington, 2004*) was used instead of *parS* (*Figure 5A*). We also failed to detect CTP binding in our DRaCALA assay or by isothermal titration calorimetry when DNA was omitted. Nevertheless, we robustly detected CTP hydrolysis to CDP and inorganic phosphate when *Caulobacter* ParB and CTP were included, albeit at a very low rate of ~0.4 CTP molecules per ParB per hour (*Figure 5C*). A background level of inorganic phosphate was observed when *Caulobacter* ParB was incubated with ATP, GTP, or UTP (*Figure 5C*). Crucially, the addition of a 22 bp *parS* DNA, but not a non-cognate 22 bp *NBS* DNA, increased CTP turnover rate sevenfold to ~3 CTP molecules per ParB per hour (*Figure 5C*). Lastly, the CTP hydrolysis was reduced to the background in the nucleation-competent but spreading-defective ParB (R104A) variant (*Figure 5C*). Altogether, our data suggest that *parS* DNA stimulates *Caulobacter* ParB to bind and hydrolyze CTP.

## ParB accumulation on DNA is unstable in the presence of a non-hydrolyzable analog CTPγS

Next, we investigated the role of CTP hydrolysis by following ParB-*parS* interaction in the presence of a non-hydrolyzable analog CTPγS (*Figure 6A* and *Figure 6—figure supplement 1*). ParB was pre-incubated with CTP or CTPγS for 1 to 60 min before binding to a 169 bp closed *parS* DNA substrate (*Figure 6A*). We observed that *Caulobacter* ParB could accumulate on DNA in the presence of CTPγS, but in contrast to when CTP was employed, a longer preincubation time between ParB and CTPγS gradually reduced ParB accumulation on DNA (*Figure 6A*). Our results suggest the possibility that CTPγS, in the absence of *parS* DNA, converts apo-ParB in solution to a nucleation-incompetent form over time. Our observation is reminiscent of a time-course experiment in which CTPγS efficiently promoted the engagement between N-terminal domains of *Bacillus* ParB in the absence of *parS* DNA (*Soh et al., 2019*). The engagement of N-terminal domains was shown to convert *Bacillus* ParB from an open to a closed protein clamp (*Soh et al., 2019*). If not already bound on DNA, the closed form of ParB presumably cannot nucleate/load onto *parS* due to its now inaccessible DNA-binding domain (*Soh et al., 2019*). We wondered if CTPγS also catalyzed the N-domain engagement in *Caulobacter* ParB in the absence of *parS* DNA. To investigate this possibility, we employed site-specific cross-linking of a purified *Caulobacter* ParB (Q35C C297S) variant by a sulfhydryl-to-sulfhydryl crosslinker bismaleimidoethane (BMOE) (*Figure 6B*). A lone cysteine residue on the native ParB was first mutated to serine to create a cysteine-less ParB (C297S) variant, then a glutamine to

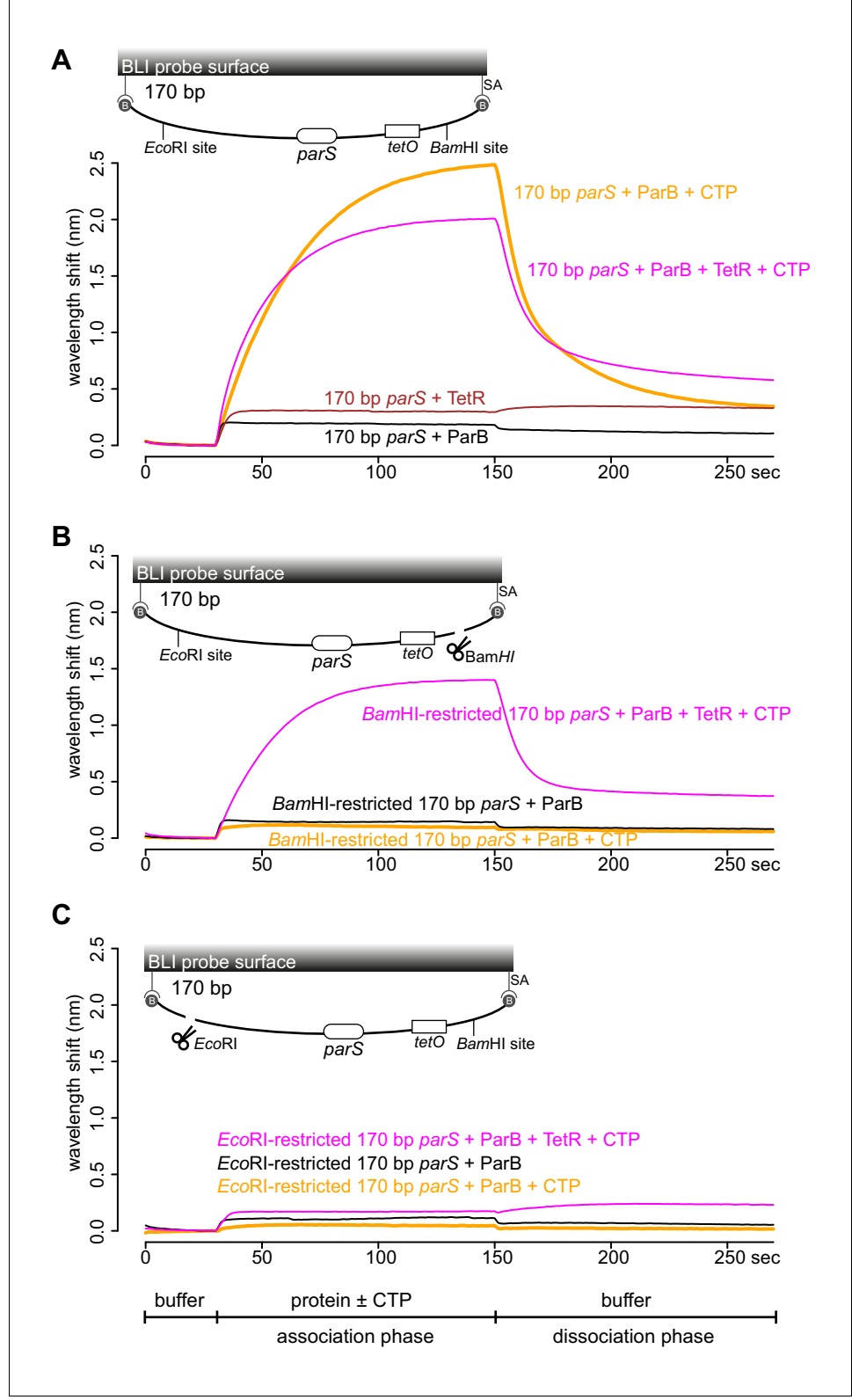

**Figure 4.** TetR-*tetO* binding restores ParB association with an open DNA substrate. (**A**) BLI analysis of the interaction between a premix of 1 μM *Caulobacter* ParB-His$_6$ ± 1 mM CTP ± 1 μM TetR-His$_6$ and a 170 bp dual biotin-labeled DNA containing a *parS* site. (**B**) Same as panel A but immobilized DNA fragments have been restricted with *Bam*HI before BLI analysis. (**C**) Same as panel A but immobilized DNA fragments have been restricted with *Eco*RI before BLI analysis. Schematic of DNA fragments together with the relative positions of *parS*,

*Figure 4 continued on next page*

*Figure 4 continued*

*tetO*, and restriction enzyme recognition sites are shown above the sensorgram. Each BLI experiment was triplicated and a representative sensorgram was presented. All buffers used for experiments in this figure contained Mg$^{2+}$.

The online version of this article includes the following source data and figure supplement(s) for figure 4:

**Source data 1.** Data used to generate *Figure 4*.

**Figure supplement 1.** Anhydrotetracycline removes DNA-bound TetR roadblock and allows ParB to accumulate further on DNA.

**Figure supplement 1—source data 1.** Data used to generate *Figure 4—figure supplement 1*.

cysteine mutation was engineered at position 35 at the N-terminal domain to create ParB (Q35C C297S). Both ParB (C297S) and ParB (Q35C C297S) were competent at spreading in the presence of CTP (*Figure 6—figure supplement 2A*). The cross-linking of ParB (Q35C C297S) was also enhanced by CTP and CTPγS but not by other NTPs, consistent with the specific role of CTP in ParB spreading (*Figure 6—figure supplement 2B*). We performed a time-course cross-linking of ParB (Q35C C297S) + CTP or CTPγS in the absence of *parS* DNA (*Figure 6B*). As shown in *Figure 6B*, CTPγS was twice as efficient as CTP in promoting the cross-linked form between N-terminal domains of *Caulobacter* ParB. The rapid increase in the nucleation-incompetent closed form of *Caulobacter* ParB might explain the overall reduction in accumulation overtime when ParB was preincubated with CTPγS (*Figure 6A*).

To further investigate the effect of CTPγS on ParB spreading on a longer time scale, we extended the association phase between 169 bp *parS* DNA and a freshly prepared premix of ParB + CTP or CTPγS from 120 s to 60 min (*Figure 6C*). In the presence of CTP, the reaction reached steady state after 120 s and remained stable for the duration of the association phase (*Figure 6C*). However, in the presence of CTPγS, ParB accumulation rapidly reached the maximal level after 200 s, then declined slowly over 60 min. We suggest that DNA-bound ParB-CTPγS complexes gradually dissociated from DNA into solution (possibly via a transient clamp opening or CTPγS leaving its weak binding pocket rather than by hydrolysis) but were not replenished by a new cycle of nucleation-spreading-dissociation because over time most ParB-CTPγS in solution was in a nucleation-incompetent closed form. Taken together, we suggest that CTP hydrolysis is not required for *parS*-bound ParB to escape the nucleation site to spread, but possibly contributes to maintaining the stability of spreading by recycling ParB.

## Discussion

In this work, we report that a small molecule (CTP) is required to enable *Caulobacter* ParB proteins (as well as eight other chromosomal ParB proteins, *Figure 2—figure supplement 3*) to spread in vitro. Recently, *Soh et al. (2019)* observed that F-plasmid and P1-plasmid ParB proteins also bind and hydrolyze CTP (*Soh et al., 2019*). Hence, it is most likely that the effect of CTP on ParB spreading is universal among plasmid and chromosomal ParB orthologs. A classical mutant whose arginine-rich patch (G$^{101}$ERRxR) has been mutated to alanine for example ParB (R104A) (*Lin and Grossman, 1998*) was not responsive to CTP, suggesting that CTP is bound to the N-terminal domain of *Caulobacter* ParB. Indeed, *Soh et al. (2019)* reported a co-crystal structure that showed CDP binding to the arginine-rich patch at the N-terminal domain of *Bacillus* ParB (CTP was hydrolyzed to CDP during crystallization) (*Soh et al., 2019*). *Osorio-Valeriano et al. (2019)* also showed a similar binding pocket of CTP at the N-terminal domain of *Myxococcus* ParB by hydrogen-deuterium exchange mass spectrometry (*Osorio-Valeriano et al., 2019*). Intriguingly, a co-crystal structure of a *Helicobacter pylori* ParB-*parS* complex, together with the in vitro magnetic-tweezer and single-molecule TIRF microscopy-based experiments with *Bacillus* ParB showed that the N-terminal domain can oligomerize to bridge DNA together without the need of an additional ligand (*Taylor et al., 2015*; *Graham et al., 2014*; *Chen et al., 2015*; *Fisher et al., 2017*). There might be two different modes of action of ParB on DNA: one for bridging DNA together (that does not require CTP) and another for the lateral spreading of ParB on DNA (that requires CTP). Investigating the relative contribution of these two different modes of action to chromosome segregation in vivo is an important challenge for the future.

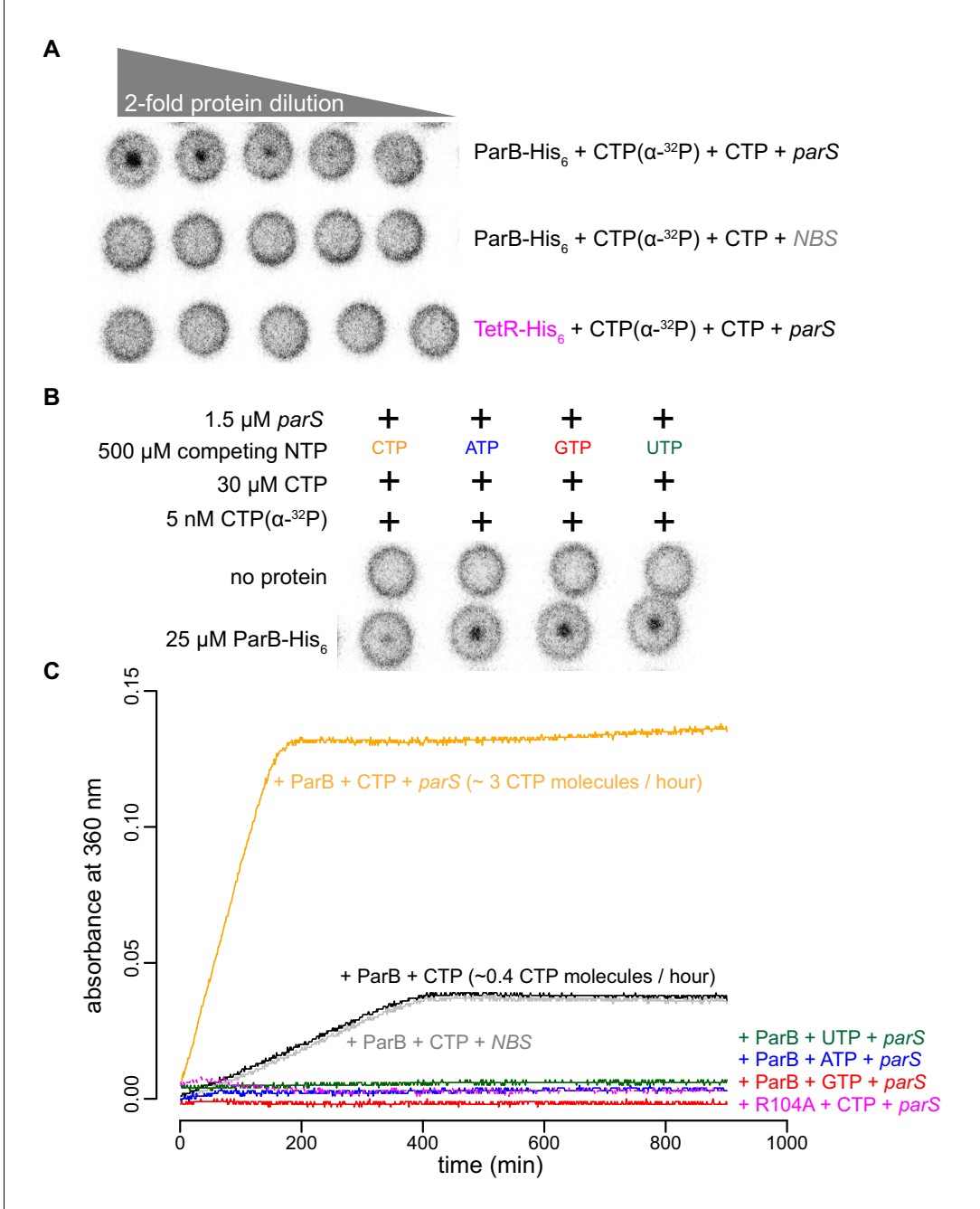

**Figure 5.** *parS* DNA increases the CTP binding and hydrolysis rate by *Caulobacter* ParB. (**A–B**) CTP binding as monitored by DRaCALA assay using radiolabeled CTP α-P[32]. The bulls-eye staining indicates CTP binding due to a more rapid immobilization of protein-ligand complexes compared to free ligands alone. The starting concentration of proteins used in panel A was 25 µM. The same concentration of radioactive CTP, unlabeled CTP, and DNA was used in experiments shown in panels A and B. (**C**) A continuous monitoring of inorganic phosphate (Pi) released by recording absorbance at 360 nm overtime at 25°C. The rates of CTP hydrolysis were inferred from a Pi standard. The NTP hydrolysis of *Caulobacter* ParB was also monitored in the presence of ATP, GTP, or UTP, with a 22 bp *parS* DNA duplex or a non-cognate 22 bp *NBS* DNA duplex (a DNA-binding site of Noc protein ***Wu and Errington, 2004***). The nucleation-competent but spreading-defective ParB (R104A) mutant did not hydrolyze CTP in the presence of *parS* DNA. All buffers used for experiments in this figure contained Mg[2+].

The online version of this article includes the following source data for figure 5:

**Source data 1.** Data used to generate ***Figure 5***.

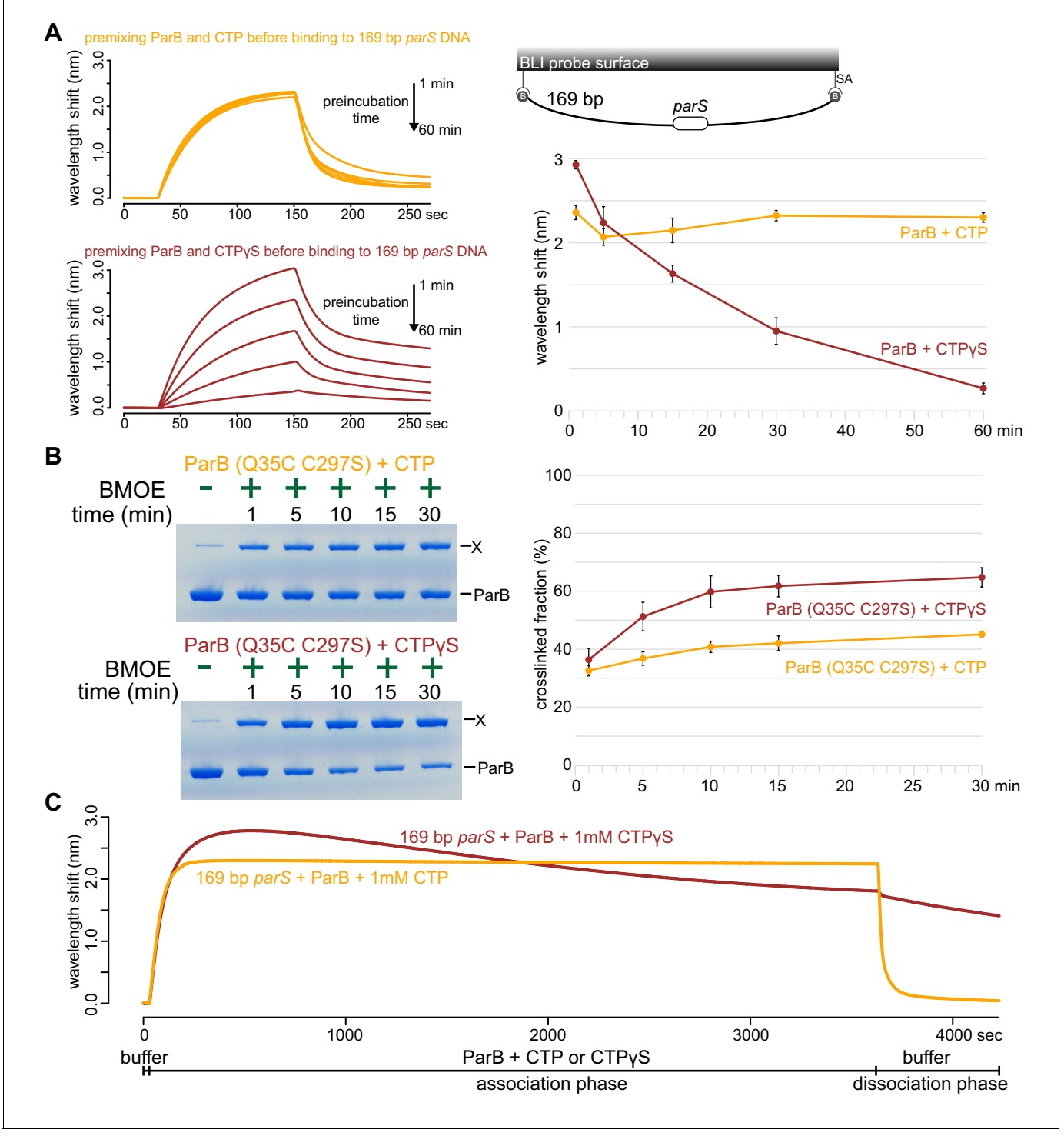

**Figure 6.** ParB accumulation on DNA is unstable in the presence of a non-hydrolyzable analog CTPγS. (**A**) (Left panel) BLI analysis of the interaction between a premix of 1 μM *Caulobacter* ParB-His$_6$ ± 1 mM CTP or CTPγS and a 169 bp dual biotin-labeled *parS* DNA. Purified ParB was preincubated with CTP or CTPγS for 1, 5, 15, 30, or 60 min before BLI analysis. (Right panel) Quantification of ParB-DNA interaction at the end of each association phase (150$^{th}$ sec). Error bars represent SD from three replicates. (**B**) (Left panel) Time course of *Caulobacter* ParB (Q35C C297S) cross-linking with CTP or CTPγS in the absence of *parS* DNA. Purified ParB-His$_6$ (Q35C C297S) (10 μM) were preincubated with 1 mM CTP or CTPγS for 1, 5, 10, 15, or 30 min before BMOE was added. X indicates a cross-linked form of ParB. (Right panel) Quantification of the cross-linked (X) fractions. Error bars represent SD from three replicates. (**C**) BLI analysis of the interaction between a premix of 1 μM *Caulobacter* ParB-His$_6$ ± 1 mM CTP or CTPγS and a 169 bp dual

*Figure 6 continued on next page*

*Figure 6 continued*

biotin-labeled *parS* DNA. CTP or CTPγS was added to purified ParB, and the mixture was immediately used for BLI experiments. All buffers used for experiments in this figure contained $Mg^{2+}$.

The online version of this article includes the following source data and figure supplement(s) for figure 6:

**Source data 1.** Data used to generate *Figure 6*.
**Figure supplement 1.** *Caulobacter* ParB binds CTP and a non-hydrolyzable analog CTPγS but not CMP-PCP.
**Figure supplement 1—source data 1.** Data used to generate *Figure 6—figure supplement 1*.
**Figure supplement 2.** BLI analysis and gel analysis of cross-linking products of purified *Caulobacter* ParB (Q35C C297S).
**Figure supplement 2—source data 1.** Data used to generate *Figure 1—figure supplement 2*.

The requirement of a DNA substrate with blocked ends for ParB accumulation in vitro is suggestive of a lateral ParB diffusion along the DNA that is ParB can escape by running off a free DNA end (*Figure 7*). Inside cells, the spreading and bridging/caging of ParB have been inferred from the compact foci of fluorescently labeled ParB (*Harms et al., 2013*; *Lagage et al., 2016*; *Sanchez et al., 2015*; *Thanbichler and Shapiro, 2006*; *Kusiak et al., 2011*; *Lin et al., 1997*; *Glaser et al., 1997*; *Erdmann et al., 1999*), presumably resulting from the concentration of fluorescent signal to a defined location in the cytoplasm. Nucleation-competent but spreading-defective ParB mutants formed no or very diffusive foci in vivo (*Graham et al., 2014*; *Song et al., 2017*). Recently, it has been observed that an artificially engineered double-strand break ~8 kb away from *parS* did not cause a dissolution of ParB foci in *Caulobacter* cells (*Badrinarayanan et al., 2015*). This result seemingly contradicted our findings that *Caulobacter* ParB spreading in vitro requires a closed DNA. However, we reason that the abundant DNA-bound transcription factors and RNA polymerases in vivo act as roadblocks to minimize ParB runoff. This barricading effect has been recapitulated in our experiments with TetR, a tight DNA-binding transcriptional regulator (*Figure 4*).

Our results so far suggest three distinct stages of *Caulobacter* ParB-DNA interactions in vitro:

Stage 1: ParB nucleates on *parS* (*Figure 7A*). Results from experiments in *Figure 1* indicate that CTP modulates ParB nucleation on a *parS* site. *Soh et al. (2019)* reported that CTP-bound ParB could form a closed protein clamp even in the absence of *parS* DNA, albeit not energetically favorable (*Soh et al., 2019*). The DNA-binding domain of a closed ParB clamp would be inaccessible to DNA, especially to a closed DNA substrate. It might be that only apo-ParB or a transiently formed CDP-bound ParB (from CTP hydrolysis) are able to nucleate on *parS* (*Figure 7A*). Supporting this interpretation, preincubation of *Caulobacter* ParB with a non-hydrolyzable analog CTPγS promoted efficiently the cross-linked form of ParB and subsequently reduced ParB nucleation and spreading on DNA (*Figure 6*). Initially, we were surprised by a weak CTP binding of *Bacillus*, *Caulobacter*, and *Myxococcus* ParBs (*Osorio-Valeriano et al., 2019*; *Soh et al., 2019*), however, this might be advantageous for the cells as a fraction of intracellular apo-ParB will remain to nucleate on *parS*.

Stage 2: Nucleated ParB escapes from *parS* (*Figure 7B–C*). We showed that *Caulobacter* ParB-*parS* complex binds CTP, and this facilitates ParB dissociation from *parS* (*Figure 1D*). It was reported that the DNA-binding domain in *Bacillus* ParB-CDP co-crystal structure is incompatible with *parS* binding (*Soh et al., 2019*) and this might enable ParB to escape from a high-affinity nucleation site to non-specific flanking DNA. Our observation of a low BLI response with an open DNA (*Figure 3* and *Figure 4*) implies that ParB proteins dissociate off the free end well before the next ParB escapes from the *parS* nucleation site. We suggest that the transition from a *parS*-bound ParB to a spreading ParB might be the rate-limiting step.

Stage 3: ParB spreads or diffuses to non-specific DNA flanking *parS*. Our observation that *Caulobacter* ParB did not accumulate on an open DNA suggests that *Caulobacter* ParB diffuses laterally along the DNA. Similarly, cross-linking experiments on *Bacillus* ParB (*Soh et al., 2019*) proposed that the ParB-CTP complex forms a sliding clamp that moves along the DNA (*Soh et al., 2019*). From our experiments with CTPγS (*Figure 6*) and consistent with *Soh et al. (2019)*, we suggest that the diffusive *Caulobacter* ParB along the DNA is CTP bound. The low CTP hydrolysis rate of *Caulobacter* ParB (~3 CTP molecules per ParB per hour) while ParB spreading could be observed by BLI within minutes also lends support to the interpretation that the diffusive spreading form of *Caulobacter* ParB is most likely CTP-bound (*Figure 7B–C*). For *Bacillus* and *Caulobacter* ParB, CTP hydrolysis is not required for ParB to escape from the nucleation site (*Soh et al., 2019*). *Caulobacter* ParB

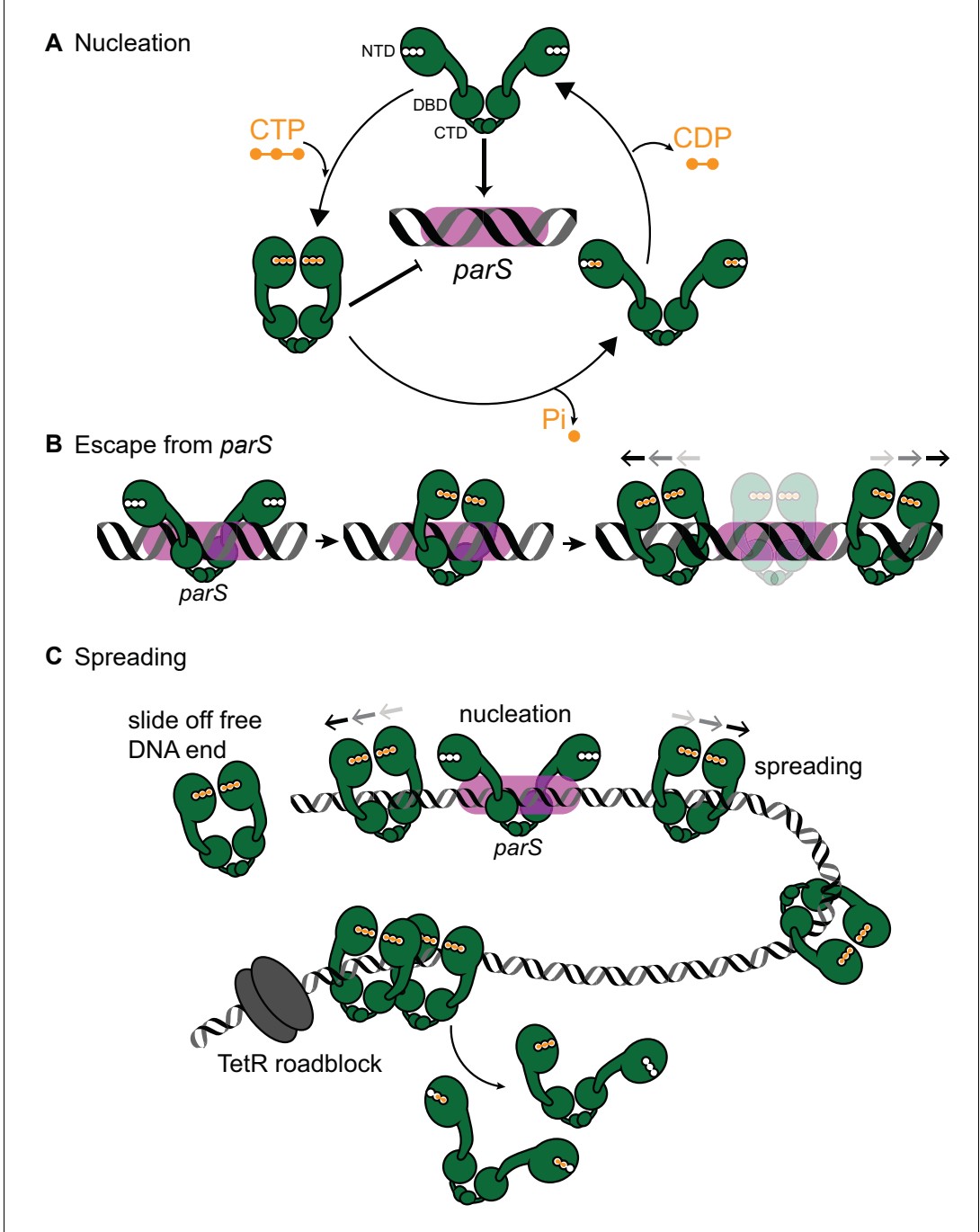

**Figure 7.** A model for *Caulobacter* ParB nucleation and spreading. (**A**) *Caulobacter* ParB nucleation at *parS*. CTP (orange) reduces *Caulobacter* ParB (dark green) nucleation at *parS* (magenta box), presumably by inducing conformational changes that are incompatible with a site-specific *parS* binding (**Soh et al., 2019**). Only apo- or CDP-bound ParB can nucleate on *parS*. ParB hydrolyzes CTP at a faster rate in the presence of *parS*. The domain architecture of ParB is also shown: NTD: N-terminal domain, DBD: DNA-binding domain, and CTD: C-terminal domain. (**B**) *Caulobacter* ParB escapes from the nucleation site *parS*. Apo-ParB at *parS* binds CTP and slides laterally away from the nucleation site *parS* while still associating with DNA. (**C**) *Caulobacter* ParB sliding and spreading on DNA. CTP-bound ParBs diffuse from the nucleation site *parS* and can run off the free DNA end unless they are blocked by DNA-bound roadblocks such as transcriptional regulators for example TetR. CTP hydrolysis is not required for ParB to escape from the nucleation *parS* site but might contribute to ParB recycling. It is not yet known whether both CTP molecules on a ParB dimer are concertedly hydrolyzed/dissociated for ParB to escape from the chromosome or a heterodimer state of ParB with a single CTP bound also exists in vivo.

could still spread on a DNA when incubated with a non-hydrolyzable CTPγS (*Figure 6*), even though spreading, at least in vitro, was less stable over time in comparison to when CTP was employed (*Figure 6C*). As suggested for *Bacillus* ParB (*Soh et al., 2019*) and supported by our results with *Caulobacter* ParB, CTP hydrolysis might contribute to ParB recycling instead. It is possible that ParB recycling in vivo is achieved via several routes: (i) CTP dissociation from its weak binding pocket, (ii) CTP hydrolysis, (iii) a possible enhanced CTP dissociation/hydrolysis via collisions with DNA-bound roadblocks, or (iv) a transient clamp opening even when ParB is CTP bound. Additional work is required to investigate the dynamics of ParB clamp opening/closing, and whether both CTP molecules on opposite subunits of a ParB dimer are concertedly hydrolyzed/dissociated for ParB to escape from the chromosome or a heterodimer state of ParB with a single CTP bound exists in vivo.

### Final perspectives

In this work, we showed the enhancing effect of CTP on *Caulobacter* ParB accumulation on DNA and further demonstrated that ParB spreading requires a closed DNA substrate and that a DNA-binding transcriptional regulator can act as a roadblock to attenuate spreading unidirectionally in vitro. Our real-time and label-free reconstitution of ParB spreading has successfully recapitulated many well-known aspects of ParB behaviors and is consistent with pioneering works by *Soh et al. (2019)* and *Osorio-Valeriano et al. (2019)*. Beyond the biological significance of the findings, our label-free approaches to biochemical reconstitution obviate the often time-consuming and challenging task of site-specifically labeling proteins with fluorophores/chemical crosslinkers without affecting the function of proteins. Here, we have demonstrated the medium-throughput capability of our methodology by investigating the effect of CTP on the spreading of eight additional chromosomal ParB proteins. The ease and medium-throughput manner of our methodology is likely to facilitate future works by the community. Important areas of research in the future are to investigate (i) the effect of ParB-CTP spreading on the supercoiling state of the DNA and vice versa, and (ii) how ParA and SMC interact with ParB-CTP in vivo to organize and faithfully segregate replicated chromosomes to each daughter cell.

# Materials and methods

## Key resources table

| Reagent type (species) or resource | Designation | Source or reference | Identifiers | Additional information |
|---|---|---|---|---|
| Strain, strain background (*Escherichia coli*) | Rosetta (DE3) | Merck | Cat# 70954 | Electro-competent cells |
| Recombinant DNA reagent | See *Supplementary file 1* | This paper | | See *Supplementary file 1* |
| Sequence-based reagent | See *Supplementary file 1* | This paper | | See *Supplementary file 1* |
| Antibody | Anti-6xHis tag antibody (HRP) (Rabbit polyclonal) | Abcam | Cat# ab1187 | Western blot (1:5,000) |
| Commercial assay or kit | EnzChek Phosphate Assay Kit | ThermoFisher | Cat# E6646 | |
| Commercial assay or kit | Gibson Assembly Master Mix | NEB | Cat# E2611S | |
| Commercial assay or kit | Gateway BP Clonase II enzyme mix | ThermoFisher | Cat# 11789020 | |
| Commercial assay or kit | Dip-and-Read Streptavidin (SA) biosensors | Molecular Devices | Cat# 18–5019 | |
| Commercial assay or kit | HisTrap High Performance column | GE Healthcare | Cat# GE17524801 | |

*Continued on next page*

*Continued*

| Reagent type (species) or resource | Designation | Source or reference | Identifiers | Additional information |
|---|---|---|---|---|
| Commercial assay or kit | HiTrap Heparin High Performance column | GE Healthcare | Cat# GE17040601 | |
| Commercial assay or kit | HiLoad 16/600 Superdex 75 pg column | GE Healthcare | Cat# GE28989333 | |
| Commercial assay or kit | Dynabeads MyOne Streptavidin C1 | ThermoFisher | Cat# 65001 | |
| Commercial assay or kit | Amersham Protran supported Western blotting membranes, nitrocellulose | GE Healthcare | Cat# GE10600016 | pore size 0.45 μm, for DRaCALA assay |
| Peptide, recombinant protein | *Bam*HI-HF | NEB | Cat# R3136S | 20,000 units/mL |
| Peptide, recombinant protein | *Bam*HI | NEB | Cat# R0136S | 20,000 units/mL |
| Peptide, recombinant protein | *Eco*RI-HF | NEB | Cat# R3101S | 20,000 units/mL |
| peptide, recombinant protein | *Hind*III-HF | NEB | Cat# R3104S | 20,000 units/mL |
| Peptide, recombinant protein | Exonuclease VII | NEB | Cat# M0379S | 10,000 units/mL |
| Peptide, recombinant protein | Exonuclease T7 | NEB | Cat# M0263S | 10,000 units/mL |
| Chemical compound, drug | Benzonase nuclease | Sigma-Aldrich | Cat# E1014 | |
| Chemical compound, drug | NTP | ThermoFisher | Cat# R0481 | 100 mM solution |
| Chemical compound, drug | CTPγS | Jena Bioscience | | A gift from S. Gruber and custom synthesis (purity ≥ 96%) |
| Chemical compound, drug | CMP-PCP | Jena Bioscience | Cat# NU-254 | |
| Chemical compound, drug | $P^{32}$-α-CTP | Perkin Elmer | Cat# BLU008H250UC | 3000 Ci/mmol, 10 mCi/ml, 250 μCi |
| Chemical compound, drug | Anhydrotetracycline hydrochloride (ahTc) | Abcam | Cat# ab145350 | Dissolved in ethanol |
| Chemical compound, drug | Bismaleimidoethane (BMOE) | ThermoFisher | Cat# 22323 | Dissolved in DMSO |
| Software, algorithm | BLItz Pro | Molecular Devices | Cat# 50–0156 | Version 1.2 https://www.moleculardevices.com/ |
| Software, algorithm | R | R Foundation for Statistical Computing | RRID: SCR_001905 | Version 3.2.4 |
| Software, algorithm | Image Studio Lite | LI-COR Biosciences | RRID: SCR_013715 | Version 5.2 |
| Software, algorithm | Excel 2016 | Microsoft | RRID: SCR_016137 | |

## Protein overexpression and purification

Full-length *Caulobacter* ParB (WT) and ParB (R104A) were purified as described previously (*Tran et al., 2018*). Briefly, pET21b::ParB-His$_6$ (WT or R104A) was introduced into *E. coli* Rosetta (DE3) competent cells (Merck). A 10 mL overnight culture was used to inoculate 4 L of LB medium + carbenicillin + chloramphenicol. Cells were grown at 37°C with shaking at 250 rpm to an OD$_{600}$ of 0.4. The culture was then left to cool to 4°C before isopropyl-β-D-thiogalactopyranoside (IPTG) was

added to a final concentration of 1 mM. The culture was shaken for 3 hr at 30℃ before cells were harvested by centrifugation.

Pelleted cells were resuspended in a buffer containing 100 mM Tris-HCl pH 8.0, 300 mM NaCl, 10 mM Imidazole, 5% (v/v) glycerol, 1 µL of Benzonase nuclease (Sigma Aldrich), 1 mg of lysozyme (Sigma Aldrich), and an EDTA-free protease inhibitor tablet (Roche). The pelleted cells were then lyzed by sonication. The cell debris was removed via centrifugation at 28,000 g for 30 min, and the supernatant was filtered through a 0.45 µm sterile filter (Sartorius Stedim). The protein was then loaded into a 1 mL HisTrap column (GE Healthcare) that had been equilibrated with buffer A [100 mM Tris-HCl pH 8.0, 300 mM NaCl, 10 mM Imidazole, and 5% glycerol]. Protein was eluted from the column using an increasing (10 mM to 500 mM) imidazole gradient in the same buffer. ParB-containing fractions were pooled and diluted to a conductivity of 16 mS/cm before being loaded onto a 1 mL Heparin HP column (GE Healthcare) that had been equilibrated with 100 mM Tris-HCl pH 8.0, 25 mM NaCl, and 5% glycerol. Protein was eluted from the Heparin column using an increasing (25 mM to 1 M NaCl) salt gradient in the same buffer. ParB that was used for EnzChek Phosphate assay and DRaCALA was further polished via a gel-filtration column. To do so, purified ParB was concentrated by centrifugation in an Amicon Ultra-15 3 kDa cut-off spin filters (Merck) before being loaded into a Superdex 75 gel filtration column (GE Healthcare). The gel filtration column was pre-equilibrated with 100 mM Tris-HCl pH 8.0, 250 mM NaCl, and 1 mM $MgCl_2$.

C-terminally His-tagged TetR (class B, from Tn10) were expressed from *E. coli* Rosetta (DE3) harboring a pET21b::TetR-His$_6$ plasmid (*Supplementary file 1*). TetR-His$_6$ were purified via a one-step Ni-affinity column using the exact buffers as employed for the purification of *Caulobacter* ParB-His$_6$.

C-terminally His-tagged ParB (C297S) and ParB (Q35C C297S) were expressed from *E. coli* Rosetta (DE3) harboring a pET21b::ParB-His$_6$ (C297S) or (Q35C C297S) plasmid (*Supplementary file 1*). ParB-His$_6$ (C297S) and (Q35C C297S) were purified via a one-step Ni-affinity column using the exact buffers as employed for the purification of *Caulobacter* ParB-His$_6$. ParB (Q35C C297S) stock solution was supplemented with TCEP (1 mM final concentration) before being flash-frozen in liquid nitrogen.

N-terminally His-tagged MBP-tagged ParB (orthologous proteins from various bacterial species) were expressed from *E. coli* Rosetta (DE3) harboring pET-His-MBP-TEV-DEST::ParB plasmids. His$_6$-MBP-ParB proteins were purified via a one-step Ni-affinity column as described previously (*Jalal et al., 2019*).

Different batches of proteins were purified by A.S.B.J and N.T.T, and are consistent in all assays used in this work. Both biological (new sample preparations from a fresh stock aliquot) and technical (same sample preparation) replicates were performed for assays described in this study.

## Construction of pET21b::ParB-His$_6$ (C297S and Q35C C297S)

DNA containing the codon-optimized coding sequence of ParB (C297S or Q35C C297S) was chemically synthesized (gBlocks dsDNA fragments, IDT). These gBlocks fragments and a *Nde*I-*Hind*III-digested pET21b backbone were assembled using a 2x Gibson master mix (NEB). Two and a half µL of each fragment at equimolar concentration was added to 5 µL 2x Gibson master mix (NEB), and the mixture was incubated at 50℃ for 60 min. Five µL was used to transform chemically competent *E. coli* DH5α cells. Gibson assembly was possible due to a 23 bp sequence shared between the *Nde*I-*Hind*III-cut pET21b backbone and the gBlocks fragments. These 23 bp regions were incorporated during the synthesis of gBlocks fragments. The resulting plasmids were sequence verified by Sanger sequencing (Eurofins, Germany).

## Construction of pET21b::TetR-His$_6$

DNA containing the coding sequence of TetR (class B, from Tn10) was chemically synthesized (gBlocks dsDNA fragments, IDT). This gBlocks fragment and a *Nde*I-*Hind*III-digested pET21b backbone were assembled together using a 2x Gibson master mix (NEB). Gibson assembly was possible due to a 23 bp sequence shared between the *Nde*I-*Hind*III-cut pET21b backbone and the gBlocks fragment. These 23 bp regions were incorporated during the synthesis of gBlocks fragments. The resulting plasmids were sequence verified by Sanger sequencing (Eurofins, Germany).

## Construction of pENTR::ParB orthologs

The coding sequences of ParB orthologs were chemically synthesized (gBlocks dsDNA fragments, IDT) and cloned into pENTR-D-TOPO backbone (Invitrogen) by Gibson assembly (NEB). The resulting plasmids were sequence verified by Sanger sequencing (Eurofins, Germany).

## Construction of pET-His-MBP-TEV-DEST::ParB orthologs

The *parB* genes were recombined into a Gateway-compatible destination vector pET-His-MBP-TEV-DEST (*Jalal et al., 2019*) via an LR recombination reaction (ThermoFisher). For LR recombination reactions: 1 µL of purified pENTR::*parB* (~100 ng/µL) was incubated with 1 µL of the destination vector pET-His-MBP-TEV-DEST (~100 ng/µL), 1 µL of LR Clonase II enzyme mix, and 2 µL of water in a total volume of 5 µL.

## Construction of DNA substrates for BLI assays

All DNA constructs (*Supplementary file 1*) were designed in VectorNTI (ThermoFisher) and were chemically synthesized (gBlocks dsDNA fragments, IDT). All linear DNA constructs were designed with M13F and M13R homologous regions at each end. To generate a dual biotin-labeled DNA substrate, PCR reactions were performed using a 2x GoTaq PCR master mix (Promega), biotin-labeled M13F and biotin-labeled M13R primers, and gBlocks fragments as template. PCR products were resolved by electrophoresis and gel purified.

## Measurement of protein-DNA interaction by bio-layer interferometry (BLI)

Bio-layer interferometry experiments were conducted using a BLItz system equipped with Dip-and-Read Streptavidin (SA) Biosensors (Molecular Devices). BLItz monitors wavelength shifts (nm) resulting from changes in the optical thickness of the sensor surface during association or dissociation of the analyte. All BLI experiments were performed at 22°C. The streptavidin biosensor was hydrated in a low-salt binding buffer [100 mM Tris-HCl pH 8.0, 100 mM NaCl, 1 mM $MgCl_2$, and 0.005% Tween 20] for at least 10 min before each experiment. Biotinylated double-stranded DNA (dsDNA) was immobilized onto the surface of the SA biosensor through a cycle of Baseline (30 s), Association (120 s), and Dissociation (120 s). Briefly, the tip of the biosensor was dipped into a binding buffer for 30 s to establish the baseline, then to 1 µM biotinylated dsDNA for 120 s, and finally to a low salt binding buffer for 120 s to allow for dissociation.

After the immobilization of DNA on the sensor, association reactions were monitored at 1 µM dimer concentration of ParB (with or without 1 µM TetR or NTPs at various concentrations) for 120 s or 60 min (*Figure 6A*). At the end of each binding step, the sensor was transferred into a protein-free binding buffer to follow the dissociation kinetics for 120 s. The sensor can be recycled by dipping in a high-salt buffer [100 mM Tris-HCl pH 8.0, 1000 mM NaCl, 1 mM $MgCl_2$, and 0.005% Tween 20] for 5 min to remove bound ParB.

For experiments where a closed DNA was cleaved to generate a free DNA end, DNA-coated tips were dipped into 300 µL of cutting solution [266 µL of water, 30 µL of 10x CutSmart buffer (NEB), and 4 µL of *Eco*RI-HF or *Bam*HI-HF restriction enzyme (20,000 units/mL)] for 30 min at 37°C.

For experiments described in *Figure 1—figure supplement 2*, $MgCl_2$ was omitted from all binding and protein storage buffers.

For experiments described in *Figure 3—figure supplement 2*, bio-layer interferometry assays were performed using 1x NEB 3.1 buffer [100 mM NaCl, 50 mM Tris-HCl pH 7.9, 10 mM $MgCl_2$, 100 µg/ml BSA] instead of the low-salt binding buffer [100 mM NaCl, 10 mM Tris-HCl pH 8.0, 1 mM $MgCl_2$, 0.005% Tween20]. After incubating 1 µM ParB and 1 mM CTP with a 169 bp *parS*-coated probe for 120 s, 4 µL of *Bam*HI (20,000 units/mL) or heat-inactivated *Bam*HI was added to a 300 µL reaction to start digesting bound DNA. The reaction was monitored for an additional 30 min. *Bam*HI was inactivated by heat at 65°C for 30 min.

For experiments described in *Figure 4—figure supplements 1*, 3 µM of anhydrotetracycline (ahTc) was used to remove bound TetR from DNA. After incubating 1 µM ParB and 1 mM CTP ±1 µM TetR with a 170 bp *parS*-coated probe for 120 s, ahTc (dissolved in ethanol) or ethanol alone was added to a 300 µL reaction to the final concentration of 3 µM or 0.01%, respectively. The reaction was monitored for an additional 120 s.

All sensorgrams recorded during BLI experiments were analyzed using the BLItz analysis software (BLItz Pro version 1.2, Molecular Devices) and replotted in R for presentation. Each experiment was triplicated, the standard deviation of triplicated sensorgrams is less than ten percent, and a representative sensorgram was presented in each figure.

To verify that dual biotin-labeled DNA fragments formed a closed substrate on the surface of the BLI probe, we performed a double digestion with Exonuclease T7 and Exonuclease VII (NEB) (*Figure 2—figure supplement 1*). DNA-coated tips were dipped into 300 µL of cutting solution [266 µL of water, 30 µL of 10x RE buffer 4 (NEB), 2 µL of exonuclease T7 (10,000 units/mL) and 2 µL of exonuclease VII (10,000 units/mL)] for 30 min at 25°C. Tips were then cut off from the plastic adaptor (*Figure 2—figure supplement 1*) and immerged into a 1x GoTaq PCR master mix [25 µL water, 25 µL 2x GoTaq master mix, 0.5 µL of 100 µM M13F oligo, and 0.5 µL of 100 µM M13R oligo]. Ten cycles of PCR were performed, and the PCR products were resolved on 2% agarose gels (*Figure 2—figure supplement 1*).

NTP (stock concentration: 100 mM) used in BLI assays was purchased from ThermoFisher. CTPγS (stock concentration: 90 mM) was a generous gift from Stephan Gruber and Young-Min Soh. CTPγS was also custom-synthesized and re-purified to 96% purity (Jena Bioscience). Another non-hydrolyzable analog CMP-PCP (Jena Bioscience) was unsuitable for our assays as *Caulobacter* ParB does not bind CMP-PCP (*Figure 6—figure supplement 1*).

## Construction of DNA substrates for pull-down assays

A 260 bp DNA fragment containing *Caulobacter parS* sites (genomic position: 4034789–4035048) (*Tran et al., 2018*) or scrambled *parS* sites were chemically synthesized (gBlocks fragments, IDT). These DNA fragments were subsequently 5' phosphorylated using T4 PNK enzyme (NEB), then cloned into a *Sma*I-cut pUC19 using T4 DNA ligase (NEB). The two resulting plasmids are pUC19::260bp-*parS* and pUC19::260bp-scrambled *parS* (*Supplementary file 1*). These plasmids were sequence verified by Sanger sequencing (Eurofins, Germany). To generate dual biotin-labeled DNA substrates, we performed PCR using a pair of biotinylated primers: around_pUC19_F and around_pUC19_R, and either pUC19::260bp-*parS* or pUC19::260bp-scrambled *parS* as a template. Phusion DNA polymerase (NEB) was employed for this round-the-horn PCR reaction. The resulting ~2.8 kb linear DNA fragments were gel-purified and eluted in 50 µL of autoclaved distilled water.

## Pull-down assays

Paramagnetic MyOne Streptavidin C1 Dyna beads (ThermoFisher) were used for pull-down assays. Thirty µL of beads were washed twice in 500 µL of high-salt wash buffer [100 mM Tris-HCl pH 8.0, 1 M NaCl, 1 mM MgCl$_2$, and 0.005% Tween 20] and once in 100 µL binding buffer [100 mM Tris-HCl pH 8.0, 100 mM NaCl, 1 mM MgCl$_2$, and 0.005% Tween 20] by repeating a cycle of resuspension and pull-down by magnetic attraction. Five µL of ~50 nM dual biotin-labeled DNA substrate was incubated with 30 µL of beads in 100 µL binding buffer for 30 min at room temperature. The reaction was occasionally mixed by pipetting up and down several times. Afterward, DNA-coated beads were washed once in 500 µL high-salt buffer [100 mM Tris-HCl pH 8.0, 1000 mM NaCl, 1 mM MgCl$_2$, and 0.005% Tween 20] and once in 500 µL of binding buffer. Finally, DNA-coated beads were resuspended in 300 µL of binding buffer. Ninety-six µL of the resuspended beads were used for each pull-down assay. Four µL of *Caulobacter* ParB-His$_6$ (WT) or (R104A) (stock concentration: 25 µM) were added to 96 µL of suspended beads. NTPs were either omitted or added to the suspended beads to the final concentration of 1 mM. The mixture was pipetted up and down several times and was left to incubate at room temperature for 5 min. Beads were then pulled down magnetically and unwanted supernatant discarded. DNA-coated beads (now with bound protein) were then washed once with 500 µL of binding buffer and once with 100 µL of the same buffer. The unwanted supernatant was discarded, and the left-over beads were resuspended in 30 µL of 1x SDS-PAGE sample buffer. Each experiment was triplicated, and a representative immunoblot was presented.

## Immunoblot analysis

For immunoblot analysis, magnetic beads were resuspended directly in 1x SDS sample buffer, then heated to 42°C for 15 min before loading to 12% Novex Tris-Glycine SDS-PAGE gels (ThermoFisher).

The eluted protein was resolved by electrophoresis at 150 V for 60 min. Resolved proteins were transferred to polyvinylidene fluoride membranes using the Trans-Blot Turbo Transfer System (Bio-Rad) and probed with 1:5000 dilution of $\alpha$-His$_6$ HRP-conjugated antibody (Abcam). Blots were imaged and analyzed using an Amersham Imager 600 (GE Healthcare) and Image Studio Lite version 5.2 (LI-COR Biosciences). The band intensities were quantified for lanes 5 and 6 (*Figure 2C*), and the range of fold difference between replicates was reported.

## Differential radial capillary action of ligand assay (DRaCALA) or membrane-spotting assay

Purified *Caulobacter* ParB-His$_6$ or TetR-His$_6$ (final concentration: 25 µM) were incubated with 5 nM radiolabeled P$^{32}$-$\alpha$-CTP (Perkin Elmer), 30 µM of unlabeled cold CTP (Thermo Fisher), 1.5 µM of 22 bp *parS* or *NBS* DNA duplex in the reaction buffer [100 mM Tris pH 8.0, 100 mM NaCl, and 10 mM MgCl$_2$] for 5 min at room temperature. For the NTP competition assay, the mixture was further supplemented with 500 µM of either unlabeled cold CTP, ATP, GTP, or UTP. Four µL of samples were spotted slowly onto a dry nitrocellulose membrane and air-dried. The nitrocellulose membrane was wrapped in cling film before being exposed to a phosphor screen (GE Healthcare) for two minutes. Each DRaCALA assay was triplicated, and a representative autoradiograph was shown.

## DNA preparation for EnzCheck phosphate assay and DRaCALA

A 22 bp palindromic single-stranded DNA fragment (*parS*: GGATGTTTCACGTGAAACATCC or *NBS*: GGATATTTCCCGGGAAATATCC) [100 µM in 1 mM Tris-HCl pH 8.0, 5 mM NaCl buffer] was heated at 98°C for 5 min before being left to cool down to room temperature overnight to form 50 µM double-stranded *parS* or *NBS* DNA. The sequences of *parS* and *NBS* are underlined.

## Measurement of NTPase activity by EnzCheck phosphate assay

NTP hydrolysis was monitored using an EnzCheck Phosphate Assay Kit (Thermo Fisher). Samples (100 µL) containing a reaction buffer supplemented with 1 mM of NTP and 1 µM ParB (WT or R104A) were assayed in a Biotek EON plate reader at 25°C for 15 hr with readings every minute. The reaction buffer (1 mL) typically contained: 740 µL Ultrapure water, 50 µL 20x customized reaction buffer [100 mM Tris pH 8.0, 2 M NaCl, and 20 mM MgCl$_2$], 200 µL MESG substrate solution, and 10 µL purine nucleoside phosphorylase (1 unit). Reactions with buffer only, buffer + protein only or buffer + NTP only were also included as controls. The plates were shaken at 280 rpm continuously for 15 hr at 25°C. The inorganic phosphate standard curve was also constructed according to the manual. Each assay was triplicated. The results were analyzed using R and the NTPase rates were calculated using a linear regression fitting in R.

## In vitro crosslinking using a sulfhydryl-to-sulfhydryl crosslinker bismaleimidoethane (BMOE)

A 50 µL mixture of 10 µM ParB (C297S) or (C297S Q35C) ± 1 mM NTP ± 0.5 µM *parS* dsDNA (22 bp) was assembled in a reaction buffer [10 mM Tris-HCl pH 7.4, 100 mM NaCl, and 1 mM MgCl$_2$] and incubated for 15 min (*Figure 6—figure supplement 2B*) or for 1, 5, 10, 15, and 30 min (*Figure 6B*) at room temperature. BMOE (1 mM final concentration from a 20 mM stock solution) was then added, and the reaction was quickly mixed by three pulses of vortexing. SDS-PAGE sample buffer containing 23 mM β-mercaptoethanol was then added immediately to quench the crosslinking reaction. Samples were heated to 50°C for 15 min before being loaded on 12% TruPAGE Tris-Glycine Precast gels (Sigma Aldrich). Each assay was triplicated. Gels were stained with InstantBlue solution (Expedeon) and band intensity was quantified using Image Studio Lite version 5.2 (LI-COR Biosciences). The crosslinked fractions were averaged, and their standard deviations calculated in Excel.

## Additional information

### Funding

| Funder | Grant reference number | Author |
| --- | --- | --- |
| Biotechnology and Biological Sciences Research Council | BB/P018165/1 | Tung BK Le |
| Royal Society | UF140053 | Tung BK Le |
| Biotechnology and Biological Sciences Research Council | BBS/E/J/000PR9791 | Ngat T Tran<br>Tung BK Le |
| Royal Society | RG150448 | Adam SB Jalal<br>Tung BK Le |
| Biotechnology and Biological Sciences Research Council | BBS/E/J/000PR9778 | Tung BK Le |

The funders had no role in study design, data collection and interpretation, or the decision to submit the work for publication.

### Author contributions
Adam SB Jalal, Ngat T Tran, Formal analysis, Investigation, Methodology; Tung BK Le, Conceptualization, Formal analysis, Supervision, Funding acquisition, Investigation, Visualization, Methodology, Writing - original draft, Writing - review and editing

### Author ORCIDs
Adam SB Jalal [iD] https://orcid.org/0000-0001-7794-8834
Ngat T Tran [iD] https://orcid.org/0000-0002-7186-3976
Tung BK Le [iD] https://orcid.org/0000-0003-4764-8851

### Decision letter and Author response
Decision letter https://doi.org/10.7554/eLife.53515.sa1
Author response https://doi.org/10.7554/eLife.53515.sa2

## Additional files

### Supplementary files
- Supplementary file 1. Plasmids, DNA, and protein sequences used in this study.

- Transparent reporting form

### Data availability
No deep sequencing data or X-ray crystallography data were generated during this study. All other data (BLI, uncropped gel images etc.) are included in the manuscript, figures, and source data files.

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
