## [Decision Letter]

**Acceptance summary:**

This work addresses a long-standing question of the mechanism by which ParB spreads onto DNA flanking the *parS* site. The paper presents a series of well-controlled and well-thought-out biochemical experiments that are some of the first to shed light on the molecular mechanism of CTP-dependent spreading by the ParB protein. Particularly, the authors biochemically differentiate *parS* binding versus ParB spreading on a variety of open or closed DNA substrates. There is a strong mechanistic focus, and the detailed biochemical dissection complements and enriches results described recently in other studies on CTP-dependent spreading by ParB.

**Decision letter after peer review:**

Thank you for submitting your article "ParB spreading on topologically closed DNA requires cytidine triphosphate in vitro" for consideration by *eLife*. Your article has been reviewed by three peer reviewers, including Anthony G Vecchiarelli as the Reviewing Editor and Reviewer #1, and the evaluation has been overseen by Gisela Storz as the Senior Editor.

The reviewers have discussed the reviews with one another and the Reviewing Editor has drafted this decision to help you prepare a revised submission.

Summary:

This well-written manuscript "ParB Spreading on topologically closed DNA requires cytidine triphosphate in vitro" by Jalal et al. addresses the long-standing question of the mechanism by which ParB spreads onto DNA flanking the *parS* site. The authors present a series of biochemical experiments that are some of the first to shed light on the molecular mechanism of CTP-dependent spreading by the ParB protein. Particularly, the authors biochemically differentiate *parS* binding versus ParB spreading on a variety of topologically open or closed DNA substrates. This work immediately follow works published by the groups of Stephan Gruber (Nov. 2019) and Martin Thanbichler (Dec. 2019) which show that ParBs from *B. subtilis* and *M. xanthus*, respectively, are CTP-binding proteins, CTPases, and CTP is required for ParB spreading. Here the authors show that *C. crescentus* ParB also shares these activities. There is a mechanistic focus on ParB spreading in the manuscript, thus the perspective is slightly different and the detailed biochemical dissection complements and enriches the results described in the other two studies. Unique to this paper, the authors show how ParBs, from a number of bacteria, require a topologically closed DNA substrate for spreading to be detected in vitro, providing strong evidence that this is likely a shared requirement amongst all ParBs. Based off the DNA binding and release kinetics and CTPase activity of ParB, a model for CTP-dependent ParB spreading is provided.

Essential revisions:

1) Reviewer 1: The final sentences of the Introduction are exclusive to those that have an intimate knowledge of the field. The authors need to spell-out their rationale for this paper given the findings of Easter and Gober, 2002; Osorio-Valeriano et al., 2019; and Soh et al., 2019. What is confirmatory and what is new in this paper compared to these recent publications. I recommend the authors use many of the points they make in the cover letter to make a more reader-friendly end to the Introduction.

2) All three reviewers independently identified several issues with the model figure that are either misleading or potentially incorrect based off the data presented in the paper. Since this is one of the first papers describing a detailed molecular mechanism for CTP-dependent ParB spreading, the authors should be incredibly careful, and specific, as to what the data has shown thus far, and what remains speculative. Refer to the reviewer comments below for details regarding specific concerns:

Reviewer 1: Figure 6 – the model makes assumptions that should be addressed in the legend or main-text. (i) CTP binding sites are shown to always react in concert with each other. Is there justification for this? It is possible that one fires and then hangs on (or releases) CDP+Pi, while the other monomer of ParB remains bound to CTP. These heterodimer states are not even considered. (ii) The model suggests only the Apo form of ParB can bridge DNA. Indeed, ParB has only been shown (by many) to bridge DNA without CTP. But the authors have not formally shown whether or not CTP-ParB shares this bridging activity – in addition to being able to spread.

Reviewer 2: The authors have convincingly demonstrated that *Caulobacter crescentus* ParB hydrolyses CTP and that *parS* stimulates the hydrolysis seven fold. However, CTP hydrolysis is not contemplated in the model. The authors state that CTP hydrolysis is not required for ParB to escape from the nucleation site and report the experiment performed with CTPγS, which show ParB spreading. If so, then what is the role of CTP hydrolysis in the system? This is not entirely clear to me. Secondly, if there is no double-strand break in the DNA, so no open end, how do the diffusing ParB dimers come off the DNA? Is the CTP hydrolysis invoked at this stage of the process maybe? But at this point, the ParB molecules are not bound to *parS*, thus the CTPase activity would be very weak. Thirdly, does ParB dimer form a completely closed ring upon CTP binding? Is it known? Overall I think that the model would benefit from some clarifications.

– Reviewers 1 and 3 independently identified critical pieces of data that, upon reviewer consultation, was deemed as necessary for supporting the author's conclusions and therefore necessary for publication:

3) Reviewer 3: The authors document *parS*-dependence of the constrained (169 bp) substrates, but not of the 20 bp substrate. Is ParB binding to the 20 bp DNA in BLI dependent on *parS*? This is an important question in part because the authors imply that binding is specific by repeatedly referring to the ParB-*parS* complexes, and also because they discuss this as a "nucleation event" at *parS*. Testing specificity is an easy experiment – measure ParB binding to the *tetO* substrates used in Figure 1C. Does CTP promote dissociation without *parS*? By the models presented, this should be confirmed.

4) Reviewer 3: Figure 1 – The data that ATP, GTP or UTP have any effect are not convincing. First, the effects are very small and second, all other ParB binding properties are unaffected by these dNTPs. The TetR control might just signal that ParB is more sensitive to salt than TetR. Also, are the counterions with the dNTPs controlled for? Most of these are sold as sodium or lithium salts (this is not mentioned in the paper). If the DNA binding is not specific for *parS*, one possibility for example, is that the non-specific DNA binding properties of ParB are more salt sensitive than its site-specific DNA binding properties. If the authors believe the effects of ATP, GTP and UTP are significant, they need to show that all salt is properly controlled. This is not a difficult experiment. It is possible that nucleotide effects would affect *parS* and non-specific DNA binding differently, and salt would need to be considered. The authors may already have done these experiments and controls, just not reported them in the paper.

5) Reviewer 1 and reviewer 3: "Spreading" – The authors are measuring DNA binding by ParB, and interpreting that this is spreading. In essence, the authors have shown *parS*-dependence if CTP is present and if the DNA is topologically tethered. This does not necessarily equate to spreading, which implies that multiple molecules of ParB associate and then slide away. The change in BLI signal with the larger substrates is consistent with this view, but a secondary confirmation that the ParB-*parS* complexes are larger would add to the confidence of the result. The authors could also be more cautious in their use of the term spreading. For example, "CTP is required for the extensive *parS*-dependent ParB spreading in vitro". How extensive is this on a 169 bp substrate? Reviewers suggest that *at least one* of the three possible experiments suggested below should be performed to support the "spreading" argument. Again, *not all* experiments below are required:

Reviewer 1: An in vitro ChIP assay to show that ParB is indeed spreading to DNA flanking the *parS* site on the topologically closed DNA substrate on the beads would be a useful addition to the interpretation of the BLI data. This is not possible with the BLI DNA because of how small the DNA fragment is. However sonication of the 2.8kb fragment down to ~250bp fragments will provide ~11 fragments in the ChIP assay with ParB antibody. This can be compared to ChIP of apo-ParB which should be sitting only on the fragments containing the *parS* site.

Reviewer 1: Not as direct as the experiment suggested above, but perhaps more feasible for the authors – Move the *tetO* site closer to and farther from the *parS* site to show a corresponding shift in the BLI shift. A closer *tetO* with TetR shown produce a lower signal. A *tetO* site with TetR bound right beside the anchor shown produce a shift similar to that of the topologically closed DNA without *tetO*.

Reviewer 3: The experiments using restriction enzyme digestion to manipulate the 169 bp substrate are a convincing way to examine the role of ends. What happens if the restriction enzyme is added *after* ParB binds? If this is feasible in the BLI context (i.e. move the probe into wells with everything plus the RE), it would be a nice way to establish that ParB slides off an end (right now it is inferred).

---

## [Author Response]

Summary:This well-written manuscript "ParB Spreading on topologically closed DNA requires cytidine triphosphate in vitro" by Jalal et al. addresses the long-standing question of the mechanism by which ParB spreads onto DNA flanking the parS site. The authors present a series of biochemical experiments that are some of the first to shed light on the molecular mechanism of CTP-dependent spreading by the ParB protein. Particularly, the authors biochemically differentiate parS binding versus ParB spreading on a variety of topologically open or closed DNA substrates. This work immediately follow works published by the groups of Stephan Gruber (Nov. 2019) and Martin Thanbichler (Dec. 2019) which show that ParBs from *B. subtilis* and M. xanthus, respectively, are CTP-binding proteins, CTPases, and CTP is required for ParB spreading. Here the authors show that C. crescentus ParB also shares these activities. There is a mechanistic focus on ParB spreading in the manuscript, thus the perspective is slightly different and the detailed biochemical dissection complements and enriches the results described in the other two studies. Unique to this paper, the authors show how ParBs, from a number of bacteria, require a topologically closed DNA substrate for spreading to be detected in vitro, providing strong evidence that this is likely a shared requirement amongst all ParBs. Based off the DNA binding and release kinetics and CTPase activity of ParB, a model for CTP-dependent ParB spreading is provided.

During the revision of this manuscript, my colleague pointed out that the term “*topologically* open/closed” might cause unintended confusion. In the original manuscript, we used the conventional definition of topology i.e. how constituent parts are linked together. However, a strict mathematical definition of DNA topology implies a specific linking number. While the conventional definition of topology has been used extensively in the literature, we agree with them that this terminology might be misinterpreted to a supercoiling-dependent DNA-binding activity by ParB-CTP in the context of our manuscript. For that reason, we replaced the term “topologically open/closed DNA” with simply “open/closed DNA” throughout the manuscript. We added sentences to the Results section to clarify this terminology:

“Immobilizing a dual biotin-labeled DNA on a streptavidin-coated BLI probe created a DNA substrate where both ends were blocked (a closed DNA) (Onn and Koshland, 2011) (Figure 2—figure supplement 1).”

“Next, we investigated whether a DNA substrate with a free end (an open DNA) can also support ParB accumulation in vitro.”

Essential revisions:1) Reviewer: The final sentences of the Introduction are exclusive to those that have an intimate knowledge of the field. The authors need to spell-out their rationale for this paper given the findings of Easter and Gober, 2002; Osorio-Valeriano et al., 2019; and Soh et al., 2019. What is confirmatory and what is new in this paper compared to these recent publications. I recommend the authors use many of the points they make in the cover letter to make a more reader-friendly end to the Introduction.

We agree and thank the reviewer for pointing this out. We have now extended the Introduction to make it more accessible to a broader audience.

2) All three reviewers independently identified several issues with the model figure that are either misleading or potentially incorrect based off the data presented in the paper. Since this is one of the first papers describing a detailed molecular mechanism for CTP-dependent ParB spreading, the authors should be incredibly careful, and specific, as to what the data has shown thus far, and what remains speculative. Refer to the reviewer comments below for details regarding specific concerns:

We agree and have either (i) toned down our interpretation of the data or (ii) performed further experiments to clarify the conclusion. Detailed responses to specific points are given below:

Reviewer 1: Figure 6 – the model makes assumptions that should be addressed in the legend or main-text. i) CTP binding sites are shown to always react in concert with each other. Is there justification for this?

We agree that there is no justification for this assumption given our current data. We have revised the Discussion and the legend of Figure 7 to make this clear. Specifically, we added the following sentence: “Additional work is also required to investigate whether both CTP molecules on opposite subunits of a ParB dimer are concertedly hydrolyzed/dissociated for ParB to escape from the chromosome or whether a heterodimer state of ParB with a single CTP bound exists in vivo.”

It is possible that one fires and then hangs on (or releases) CDP+Pi, while the other monomer of ParB remains bound to CTP. These heterodimer states are not even considered.

We agree that our current data do not rule out the absence/existence of the heterodimer state. We have now revised the Discussion and the legend of Figure 7 to make this clear to readers (see above).

ii) The model suggests only the Apo form of ParB can bridge DNA. Indeed, ParB has only been shown (by many) to bridge DNA without CTP. But, the authors have not formally shown whether or not CTP-ParB shares this bridging activity…in addition to being able to spread.

We indeed have not investigated the bridging activity of *Caulobacter* ParB in this manuscript. This is an important point and warrants an in-depth investigation in future studies. We have now removed the cartoon representation of bridging from the model in Figure 7C. We revised the Discussion and added the following sentence: “There might be two different modes of action of ParB on DNA: one for bridging DNA together (that does not require CTP) and another for the lateral spreading of ParB on DNA (that requires CTP). Investigating the relative contribution of these two different modes of action to chromosome segregation in vivo is an important challenge for the future”.

Reviewer 2: The authors have convincingly demonstrated that Caulobacter crescentus ParB hydrolyses CTP and that parS stimulates the hydrolysis seven fold. However, CTP hydrolysis is not contemplated in the model. The authors state that CTP hydrolysis is not required for ParB to escape from the nucleation site and report the experiment performed with CTPγS, which show ParB spreading. If so, then what is the role of CTP hydrolysis in the system? This is not entirely clear to me. Secondly, if there is no double-strand break in the DNA, so no open end, how do the diffusing ParB dimers come off the DNA? Is the CTP hydrolysis invoked at this stage of the process maybe? But at this point, the ParB molecules are not bound to parS, thus the CTPase activity would be very weak. Thirdly, does ParB dimer form a completely closed ring upon CTP binding? Is it known? Overall I think that the model would benefit from some clarifications.

The reviewer raised important points and we felt it is prudent to perform further experiments with CTPγS to clarify the model:

Experiment 1 (Figure 6A): We fortuitously discovered that a long preincubation of purified ParB and CTPγS (but not CTP) had a negative effect on ParB accumulation on a closed DNA substrate. To investigate further, ParB was preincubated with CTP or CTPγS for 1, 5, 15, 30, or 60 min before binding to a 169-bp closed DNA substrate (Figure 6A). We observed that *Caulobacter* ParB could accumulate on DNA in the presence of CTPγS, but in contrast to when CTP was employed, a longer preincubation time between ParB and CTPγS gradually reduced ParB accumulation on DNA (Figure 6A). Our results suggest the possibility that CTPγS, in the absence of *parS* DNA, gradually converts apo-ParB in solution to a nucleation-incompetent form over time. This observation is reminiscent of a time-course experiment in which CTPγS efficiently promoted the engagement between N-terminal domains of *Bacillus* ParB in the absence of *parS* DNA (Soh et al., 2019). The engagement of N-terminal domains was shown to convert *Bacillus* ParB from an open to a closed protein clamp (Soh et al., 2019). If not already bound on DNA, the closed form of ParB presumably cannot self-nucleate/load onto *parS* due to its now inaccessible DNA-binding domain (Soh et al., 2019).

Experiment 2 (Figure 6B): We wondered if CTPγS also catalyzes the N-domain engagement in *Caulobacter* ParB in the absence of *parS* DNA. To investigate this possibility, we employed site-specific cross-linking of a purified *Caulobacter* ParB (Q35C C297S) variant by a sulfhydryl-to-sulfhydryl crosslinker bismaleimidoethane (BMOE) (Figure 6B, same strategy as pioneered by the Gruber lab; Soh et al., 2019). We performed controls to confirm that Q35C and C297S mutations did not impair ParB activity in vitro (Figure 6—figure supplement 2). We then performed a time-course cross-linking of ParB (Q35C C297S) + CTP or CTPγS in the absence of *parS* DNA (Figure 6B). CTPγS was twice as efficient as CTP in promoting the cross-linked form between N-terminal domains of *Caulobacter* ParB (Figure 6B). The rapid increase in the nucleation-incompetent closed form of *Caulobacter* ParB might explain the overall reduction in spreading over time when ParB was preincubated with CTPγS (Figure 6A).

Experiment 3 (Figure 6C): To further investigate the effect of CTPγS on ParB spreading on a longer time scale, we extended the association phase between a 169-bp *parS* DNA and a freshly prepared premix of ParB + CTP or CTPγS from 120 sec to 60 min (Figure 6C). In the presence of CTP, the reaction reached a steady state after 120 sec and remained stable for the duration of the association phase (Figure 6C). However, in the presence of CTPγS, ParB accumulation rapidly reached the maximal level after 200 sec, then declined slowly over the course of 60 min. We reason that DNA-bound ParB-CTPγS complexes gradually dissociated from DNA into solution but were not replenished by a new cycle of nucleation-spreading-dissociation because over time most ParB-CTPγS in solution was in a nucleation-incompetent closed form. ParB-CTPγS could possibly escape the DNA via transient ring opening or CTPγS leaving its weak binding pocket rather than by hydrolysis.

Taken together, we suggest that CTP hydrolysis is *not* required for *parS*-bound ParB to escape the nucleation site *parS* to spread, but possibly contributes to the stability of spreading by recycling ParB.

We hope these new data shed further light on the possible role of CTP hydrolysis and answer some of the questions that the reviewer asked. Nevertheless, we have been cautious with the interpretation in the revised manuscript and only suggest that “CTP hydrolysis possibly contributes to ParB recycling”. We also refrained from saying “ParB forms a completely closed ring upon CTP binding”, since the dynamics of ring opening and closing (when CTP is bound) are not yet known. Specifically, we added the following sentences to the Discussion:

“It is possible that ParB recycling in vivo is achieved via several routes: (i) CTP dissociation from its weak binding pocket, (ii) CTP hydrolysis, (iii) a possible enhanced CTP dissociation/hydrolysis via collisions with DNA-bound roadblocks, or (iv) a transient clamp opening even when ParB is CTP bound. Additional work is required to investigate the dynamics of ParB clamp opening/closing, and whether both CTP molecules on opposite subunits of a ParB dimer are concertedly hydrolyzed/dissociated for ParB to escape from the chromosome or a heterodimer state of ParB with a single CTP bound exists in vivo.”

– Reviewers 1 and 3 independently identified critical pieces of data that, upon reviewer consultation, was deemed as necessary for supporting the author's conclusions and therefore necessary for publication. All three reviewers think the experiments suggested below can be done in 2 months:3) Reviewer 3: The authors document parS-dependence of the constrained (169 bp) substrates, but not of the 20 bp substrate. Is ParB binding to the 20 bp DNA in BLI dependent on parS? This is an important question in part because the authors imply that binding is specific by repeatedly referring to the ParB-parS complexes, and also because they discuss this as a "nucleation event" at parS. Testing specificity is an easy experiment – measure ParB binding to the tetO substrates used in Figure 1C. Does CTP promote dissociation without parS? By the models presented, this should be confirmed.

Unlike *Bacillus* ParB, which has an additional non-specific DNA-binding activity at its C-terminal domain (Taylor et al., 2015, Graham et al., 2015, Fisher et al., 2019), the C-terminal domain of *Caulobacter* ParB lacks this non-specific DNA-binding activity (Tran et al., 2018). Therefore, *Caulobacter* ParB distinguishes well between *parS* and non-specific DNA via its central DNA-binding domain (Tran et al., 2018). In the context of this manuscript, we have performed the suggested experiment; we measured the binding of a range of concentrations of purified *Caulobacter* ParB with 20-bp *parS* or 28-bp *tetO* and between TetR and *parS* or *tetO* DNA (Figure 1—figure supplement 1). Consistent with the literature, *Caulobacter* ParB and TetR are specific to their cognate binding sites. We have added these results to Figure 1—figure supplement 1. Finally, because *Caulobacter* ParB does not bind non-*parS* DNA to begin with, we cannot test whether CTP promotes dissociation without *parS*.

4) Reviewer 3: Figure 1 – The data that ATP, GTP or UTP have any effect are not convincing. First, the effects are very small and second, all other ParB binding properties are unaffected by these dNTPs. The TetR control might just signal that ParB is more sensitive to salt than TetR. Also, are the counterions with the dNTPs controlled for? Most of these are sold as sodium or lithium salts (this is not mentioned in the paper). If the DNA binding is not specific for parS, one possibility for example, is that the non-specific DNA binding properties of ParB are more salt sensitive than its site-specific DNA binding properties. If the authors believe the effects of ATP, GTP and UTP are significant, they need to show that all salt is properly controlled. This is not a difficult experiment. It is possible that nucleotide effects would affect parS and non-specific DNA binding differently, and salt would need to be considered. The authors may already have done these experiments and controls, just not reported them in the paper.

We agree with the reviewer that the effect of ATP, GTP, and UTP was small and did not affect other binding properties of *Caulobacter* ParB. We further performed BMOE cross-linking experiment to investigate the effect of NTPs on the N-domain engagement of *Caulobacter* ParB, and indeed ATP, GTP, and UTP had a negligible effect (Figure 6—figure supplement 2B).

We have also performed additional experiment to investigate the sensitivity of ParB to salts/counterions (Na^+^)/other contaminants that came with the purchased NTP solutions. Specifically, we measured binding of ParB to a 20-bp *parS* DNA with/without NTPs in buffer lacking Mg^2+^. All NTPs (in buffer lacking Mg^2+^) decreased ParB-*parS* interaction very slightly (but reproducibly), suggesting that ParB might be slightly sensitive to salt/counterions/contaminants in the purchased NTP solutions. Again, given that the effect of ATP, GTP, or UTP (in buffer with or without Mg^2+^) in all different types of experiments (bio-layer interferometry/BMOE cross-linking/DRaCALA/NTP hydrolysis) was small, we toned down our interpretation and focused solely on the specific effect of CTP on *Caulobacter* ParB properties.

5) Reviewer 1 and reviewer 3: "Spreading" – The authors are measuring DNA binding by ParB and interpreting that this is spreading. In essence, the authors have shown parS-dependence if CTP is present and if the DNA is topologically tethered. This does not necessarily equate to spreading, which implies that multiple molecules of ParB associate and then slide away. The change in BLI signal with the larger substrates is consistent with this view, but a secondary confirmation that the ParB-parS complexes are larger would add to the confidence of the result. The authors could also be more cautious in their use of the term spreading. For example, "CTP is required for the extensive parS-dependent ParB spreading in vitro". How extensive is this on a 169 bp substrate?

These are excellent points and we agree with the reviewers. In the revised manuscript, we are more cautious in using the term “spreading”. Instead, we use the phrase “accumulation of ParB” whenever appropriate and only use “spreading” when data are available to back up this claim.

Reviewers suggest that at least one of the three possible experiments suggested below should be performed to support the "spreading" argument. Again, not all experiments below are required:

Clear instruction from the editor and reviewers here is much appreciated. Below are the results of the suggested experiments:

Reviewer 3: The experiments using restriction enzyme digestion to manipulate the 169 bp substrate are a convincing way to examine the role of ends. What happens if the restriction enzyme is added after ParB binds? If this is feasible in the BLI context (i.e. move the probe into wells with everything plus the RE), it would be a nice way to establish that ParB slides off an end (right now it is inferred).

We have now performed this experiment with BamHI (or heat-killed BamHI as a negative control) added after ParB-CTP was bound on a closed DNA substrate (Figure 3—figure supplement 2). After adding BamHI, ParB binding to DNA reduced gradually over the course of 30 minutes, while it was unaffected when an inactivated BamHI was used instead (Figure 3—figure supplement 2). This result backs up our interpretation that ParB slides off an open DNA end.

N.B. Experiments were performed at 22^o^C instead of at 37^o^C to keep ParB stable. Also, the binding of ParB-CTP to DNA was performed in NEB buffer 3.1 [100 mM NaCl, 50 mM Tris-HCl, 10 mM MgCl2, 100 µg/ml BSA, pH 7.9] instead of the low-salt binding buffer used throughout the manuscript [100 mM NaCl, 10 mM Tris-HCl, 1 mM MgCl2, 0.005% Tween20, pH 8.0]. NEB buffer 3.1 was used to ensure BamHI digested DNA as optimally as possible while keeping *Caulobacter* ParB stable. Due to the change in buffers, the BLI response of ParB-CTP with DNA was higher in buffer NEB 3.1 than in the usual binding buffer used throughout the manuscript. Materials and methods section has been updated to reflect this.

Reviewer: Not as direct as the experiment suggested above, but perhaps more feasible for the authors – Move the tetO site closer to and farther from the parS site to show a corresponding shift in the BLI shift. A closer tetO with TetR shown produce a lower signal. A tetO site with TetR bound right beside the anchor shown produce a shift similar to that of the topologically closed DNA without tetO.

We performed a variant of the suggested experiment (Figure 4—figure supplement 1): a premix of ParB, CTP, and TetR was allowed to bind to a closedDNA substrate and, after the system reached steady state, anhydrotetracycline was added to remove bound TetR from the DNA. After the roadblock (TetR) was removed by anhydrotetracycline, we observed that ParB binding to the closed DNA increased (dotted magenta line, Figure 4—figure supplement 1), consistent with the “spreading” phenomenon. We have performed a control showing that ParB-CTP binding to DNA is insensitive to added anhydrotetracycline in the absence of TetR (solid and dotted orange lines, Figure 4—figure supplement 1).

Reviewer 1: An in vitro ChIP assay to show that ParB is indeed spreading to DNA flanking the parS site on the topologically closed DNA substrate on the beads would be a useful addition to the interpretation of the BLI data. This is not possible with the BLI DNA because of how small the DNA fragment is. However sonication of the 2.8kb fragment down to ~250bp fragments will provide ~11 fragments in the ChIP assay with ParB antibody. This can be compared to ChIP of apo-ParB which should be sitting only on the fragments containing the parS site.

We attempted this experiment for the original manuscript but were not successful, most likely because fixing *Caulobacter* ParB to DNA (+/- CTP) in vitro with 1% formaldehyde inactivated most of ParB protein. We have not optimized the concentration of formaldehyde for fixation any further due to time constraints. In previous work, we were successful in performing a non-crosslinking version of in vitro ChIP-seq (IDAP-seq) using sonicated genomic DNA and purified ParB (Tran et al., 2018). However, this methodology is inappropriate in the presence of CTP since we now have ample evidence that ParB-CTP slides off the ends of sonicated DNA.

N.B.While assembling source data to upload to *eLife*, I found that some parts of Figure 2—figure supplement 3 were mislabeled in the original manuscript. Specifically, *Morella thermoacetica* His_6_-MBP-ParB should be *Dechloromonas aromatica* His_6_-MBP-ParB, and 0.01 mM CTP was used for *Zymomonas mobilis* and *Xanthomonas campestris* ParB instead of 1mM CTP used for all other ParB samples. To double check, we purified these eight proteins again using batch purification and redid the experiments for Figure 2—figure supplement 3. I apologize for the oversight. These two mislabellings have been corrected and the figure legend updated in the revised manuscript. The conclusion that “we performed BLI experiments for eight additional chromosomal ParB proteins from a diverse set of bacterial species and consistently observed the specific effect of CTP on enhancing ParB association with a closed DNA in vitro (Figure 2—figure supplement 3). It is most likely that ParB-CTP interaction with DNA is conserved among ParB orthologs”stands.

A higher concentration of CTP was inhibitory to *Zymomonas* and *Xanthomonas* ParB, presumably because most of these apo-ParB was converted to the closed clamp form before exposure to tested DNA.